# Multi-objective Deep Data Generation with Correlated Property Control

**Shiyu Wang**[1], **Xiaojie Guo**[2], **Xuanyang Lin**[1], **Bo Pan**[1], **Yuanqi Du**[3], **Yinkai Wang**[4], **Yanfang Ye**[5],
**Ashley Ann Petersen**[6], **Austin Leitgeb**[6], **Saleh AlKhalifa**[7], **Kevin Minbiole**[6], **William Wuest**[1],
**Amarda Shehu**[8], **Liang Zhao**[1,†]

[1]Emory University, {shiyu.wang, mike.lin, bo.pan, william.wuest, liang.zhao}@emory.edu
[2]IBM Thomas.J. Watson Research Center, xguo7@gmu.edu
[3]Cornell University, yd392@cornell.edu
[4]Tufts University, yinkai.wang@tufts.edu
[5]University of Notre Dame, yye7@nd.edu
[6]Villanova University, {apeter24, austin.leitgeb, kevin.minbiole}@villanova.edu
[7]Recursiv LLC, salehesam@gmail.com
[8]George Mason University, ashehu@gmu.edu

## Abstract

Developing deep generative models has been an emerging field due to the ability to model and generate complex data for various purposes, such as image synthesis and molecular design. However, the advancement of deep generative models is limited by challenges to generate objects that possess multiple desired properties: 1) the existence of complex correlation among real-world properties is common but hard to identify; 2) controlling individual property enforces an implicit partially control of its correlated properties, which is difficult to model; 3) controlling multiple properties under various manners simultaneously is hard and under-explored. We address these challenges by proposing a novel deep generative framework, CorrVAE, that recovers semantics and the correlation of properties through disentangled latent vectors. The correlation is handled via an explainable mask pooling layer, and properties are precisely retained by generated objects via the mutual dependence between latent vectors and properties. Our generative model preserves properties of interest while handling correlation and conflicts of properties under a multi-objective optimization framework. The experiments demonstrate our model's superior performance in generating data with desired properties. The code of CorrVAE is available at `https://github.com/shi-yu-wang/CorrVAE`.

## 1 Introduction

Developing powerful deep generative models has been an emerging field due to its capability to model and generate high-dimensional complex data for various purposes, such as image synthesis [4, 30], molecular design [24, 47, 9], protein design [14, 16], co-authorship network analysis [6] and natural language generation [22, 32]. Extensive efforts have been spent on learning underlying low-dimensional representation and the generation process of high-dimensional data through deep generative models such as variational autoencoders (VAE) [27, 35, 9], generative adversarial networks (GANs) [11, 12], normalizing flows [40, 5], etc [48, 17, 8]. Particularly, enhancing the disentanglement and independence of latent dimensions has been attracting the attention of the community [4, 43, 3, 34, 45, 23], enabling controllable generation that generates data with desired properties by interpolating latent variables [44, 13, 29, 25, 38, 20, 7, 49]. For instance, CSVAE

---

† Corresponding author.

36th Conference on Neural Information Processing Systems (NeurIPS 2022).

transfers image attributes by correlating latent variables with desired properties [28]. Semi-VAE pairs latent space with properties by minimizing the mean-square-error (MSE) between latent variables and desired properties [31]. Property-controllable VAE (PCVAE) synthesizes image objects with desired positions and scales [13] by enforcing the mutual dependence between disentangled latent variables and properties. Conditional Transformer Language (CTRL) model generates text with task-specific style and contents [26]. Despite of the rapid growth of research regarding property

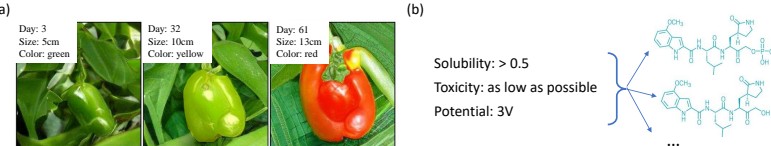

Figure 1: (a) Correlated properties are common for the real-world object, such as the day (growth time), size and color of wild pepper; (b) Generation of molecules that satisfy desired properties can be viewed as a multi-objective optimization task

controllable generation in various domains, critical challenges still remain such as: 1) **Difficulty in identifying property correlation**. Existing models for data property control typically map each property to its exclusive latent variables. Also, all the latent variables are inherently enforced to be independent of each other. Therefore such complete distentanglement among the consideration of properties disallows the model to characterize the correlation among properties and hence can only work for controlling properties that are independent of each other. However, properties in real data objects are usually correlated (Figure 1 (a)). For example, in a human face image, the face width has a correlation with eye size. The color of the wild pepper has a correlation with the growth time (Figure 1 (a)). The correlation of properties has been under-explored, which largely impairs the effectiveness of generative models; 2) **Controlling individual property also enforces an implicit partial control of its correlated properties, which is difficult to model.** For those correlated properties, controlling one of them will also constrain the others into some subspace such as a hyperplane or even a non-convex set. For example, when generating face images, if we constrain the width of the face is 100 pixels then the size of the eyes will be constrained to a reasonable range; 3) **Difficulty in simultaneously controlling multiple properties under various manners**. The real-world application usually requires generated object to satisfy multiple constraints of properties simultaneously. One may want to maximize a property's value, fix another property to a certain value, and constrain the third property within a range. Therefore, the data generation problem is entangled with and hardened by the multi-objective optimization goal, which has not been well explored. For example, chemists may design a molecule that has specific potential, minimizes its toxicity and meanwhile possesses solubility within a range (Figure 1 (b)). We overcome these challenges by proposing a novel deep generative model, CorrVAE, that recovers semantics and the correlation of properties via disentangled latent vectors. The correlation is handled by an explainable mask pooling layer, and properties are precisely retained by the generated data via the mutual dependence between latent vectors and properties. Our generative model preserves multiple properties of interest while handling correlation and conflicts of properties under a multi-objective optimization framework. The contributions of this paper are summarized as follows:

- **A novel deep generative model for multi-objective control of correlated properties.** Beyond disentangled representation learning, we aim at corresponding latent variables to target properties for better interpretability and controllability. The model is generic to different types of data such as images and graphs, together with disentangled terms to obtain independent latent variables to jointly handle correlated properties.

- **A correlated invertible mapping is proposed for mapping correlated real properties to latent independent variables.** An interpretable mask pooling layer has been proposed to explicitly identify how the real-world properties are generated by the corresponding subsets of latent independent variables. The information of these latent variables will be aggregated and enforced to be mutually dependent on the property via an invertible constraint.

- **A multi-objective optimization framework is formulated for deep data generation problem**. Corresponding latent variables in the low-dimensional representation are optimized under multiple objectives and constraints for property control purposes. Our framework is generic to various multiple objectives such as optimizing a property value, constraining property values into a range, maximizing or minimizing a property value while maintaining the correlation among properties.

- **Extensive experiments are conducted on real-world datasets.** The proposed model can generate data with multiple desired properties simultaneously in the generation process, demonstrating the effectiveness of the model. Moreover, our model shows superior accuracy of generated properties against target properties compared with comparison models with multiple real-world datasets.

This paper firstly introduces the general framework of the proposed model. Then we will discuss the details of the model, including the derivation of the overall objective, the mask pooling layer, and invertible mapping between latent space and correlated properties. Lastly, we conduct the comprehensive experiment to compare our model with existing methods.

## 2 Related works

### 2.1 Disentangled representation learning

Disentangled representation learning aims to encode information of high-dimensional complex data into a low-dimensional space that consists of mutually independent variables to separate out independent factors of variation of data distribution in the representation [43, 1, 3, 34, 21, 19, 10]. Because of the success of VAE and GANs as deep generative models [35, 18, 36, 15], a few techniques have been developed as variations of VAE or GANs to achieve disentanglement of latent variables in the representation space. For instance, $\beta$-VAE modifies the variational evidence lower bound (ELBO) by adding a hyperparameter $\beta$ before the KL-divergence term to encourage disentangled latent variables [21]. Instead, cycle-consistent VAE was proposed for supervised disentanglement of specified and unspecified factors of variation using pairwise similarity labels [23]. On the other hand, InfoGAN maximizes the mutual information between the latent variable and the generated sample under the framework of GANs. Disentangled Representation learning-Generative Adversarial Network (DR-GAN) disentangles the representation with the pose of the face on the image through the pose code provided to the generator and pose estimation by the discriminator [43].

### 2.2 Property controllable deep generative models

Considerable efforts have been spent on developing deep generative models that generate data with desired properties [13, 47, 20, 26, 25, 42, 37]. Techniques for property controllable generation include but are not limited to 1) reinforcement learning (RL) approach to goal-directed data generation that preserves target properties, such as Graph Convolutional Policy Network (GCPN) [47] and GraphAF [41], and 2) mutual dependence between properties and latent variables to control generation process by manipulating values of latent variables, such as PCVAE [13] and Conditional Subspace VAE (CSVAE) [28]. RL approach nevertheless suffers from the requirement of a large sample size for the training purpose. Moreover, all the above methods are unable to precisely capture complex correlation among properties in an explainable way, nor can they generate data that simultaneously satisfy multiple correlated targets of either values or ranges. To fill the gap between the existing methods and the need for controllable generation from various domains, we propose a novel controllable deep generative model that handles the correlation of properties via an explainable mask pooling layer and generates data with desired properties under a multi-objective optimization framework.

## 3 Problem formulation

Suppose we have a dataset $\mathcal{D}$, in which each sample can be represented as $x$, along with $y = \{y_1, y_2, ..., y_m\}$ as $m$ properties of $x$, which can be either correlated or independent with each other. For instance, if the data is a molecule, properties can be molecular weight, polarity or solubility. We further assume that $(x, y)$ is generated via some random processes from continuous latent variables in $(w, z)$, where $w$ controls the properties of interest in $y$ and $z$ controls all other aspects of $x$.

We aim to learn a generative model that generates $(x, y)$ conditioning on $(w, z)$, where $z$ is disentangled with $w$ and variables in $w$ are disentangled with each other to control either correlated or independent properties. Once the model is trained, the user can generate data with target values or ranges of properties via editing the corresponding elements in $w$. For example, we may want to generate a molecule with a specific value of the weight, and solubility within a range while minimizing its toxicity by changing values of $w$ that contribute to those properties. This goal leads to the following questions answered by our work: how to automatically identify the correlation

among properties, how to control individual property while enforcing an implicit partial control of its correlated properties, and how to control multiple properties simultaneously?

## 4 Proposed approach

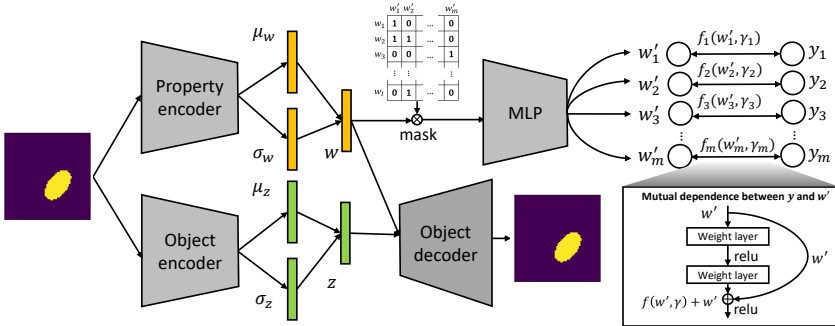

Figure 2: Overal framework of CorrVAE. CorrVAE encodes the information of correlated properties into the latent space $w$ and other information of the object into $z$ via the property and the object encoder, respectively. Then the correlation among properties is captured by the mask pooling layer, where the information to predict a specific property is aggregated into the bridging latent variable $w'$. The mutual dependence between $w'$ and the corresponding property is enforced by the invertible constraint achieved by ResNet. Lastly, the data can be generated from $(w, z)$ via the object decoder.

In general, the proposed approach, CorrVAE, identifies the property correlation via a novel mask pooling layer. It precisely retains properties via the constraint of mutual dependence between latent vectors and properties, and simultaneously controls multiple properties under a multi-objective optimization framework. The overall framework of the model is shown in Figure 2. Specifically, the model contains two phases: (1) **Learning phase** encodes the information of properties via the property encoder and other information of the data via the object encoder. As shown in Figure 2, the correlation information of properties is captured by a novel mask pooling layer. The mutual dependence between latent variables and properties is enforced via a constraint applied to the learning objective. The data is generated via the object decoder in Figure 2; (2) **Generation phase** generates data with desired properties of specific values or within target ranges under the multi-objective optimization framework. In this section, we will introduce two phases in detail.

### 4.1 Learning phase

#### 4.1.1 Overall objective for disentangled learning on latent variables

The goal requires us to not only model the dependence between $x$ and $(w, z)$ for latent representation learning and data generation, but also model the dependence between $y$ and $w$ for controlling the property. We propose to achieve this by maximizing the joint log likelihood $p(x, y)$ via its variational lower bound. Given an approximate posterior $q(z, w|x, y)$, we can use the Jensen's equality to obtain the variational lower bound of $\log p(x, y)$ as:

$$\log p(x, y) = \log \mathbb{E}_{q(z, w|x, y)}[p(x, y, w, z)/q(z, w|x, y)] \geq \mathbb{E}_{q(z, w|x, y)}[\log p(x, y, w, z)/q(z, w|x, y)]. \quad (1)$$

The joint likelihood $\log p(x, y, w, z)$ is further decomposed as $\log p(x, y|z, w) + \log p(z, w)$ given Two assumptions: (1) $x$ and $y$ are conditionally independent given $w$ since $w$ only captures information from $y$; (2) $z$ is independent from $w$ and $y$, equivalent to $y \perp z|w$. This gives us $x \perp y|(w, z)$, suggesting that $\log p(x, y|w, z) = \log p(x|w, z) + \log p(y|w, z) = \log p(x|w, z) + \log p(y|w)$. Consequently, we write the joint log-likelihood and maximize its lower bound:

$$\log p_{\theta, \gamma}(x, y, w, z) = \log p_\theta(x|w, z) + \log p(w, z) + \log p_\gamma(y|w)$$

$$= \log p_\theta(x|w, z) + \log p(w, z) + \sum_{i=1}^{m} \log p_\gamma(y_i|w'_i), \quad (2)$$

where we define a next-level latent variable $w'_i$ as the set of values in $w$ that are independent with each other and contribute to the $i$-th property to bridge the mapping $w \rightarrow y$ while allowing property controlling. Each value $w'_i$ in $w' = \{w'_1, w'_2, ..., w'_m\}$ relates to each $y_i$ in $y$. Since properties are

independent conditioning on $w'$, the decomposition of the third term holds in Eq. 2. The relationship between $y$ and $w'$ will be further explained in Section 4.1.2. Given $q_\phi(w, z|x, y) = q_\phi(w, z|x) = q_\phi(w|x) \cdot q_\phi(z|x)$, we rewrite the joint probability in Eq. (2) as the form of the Bayesian variational inference as the first term of the learning objective:

$$\mathcal{L}_1 = -\mathbb{E}_{q_\phi(w,z|x)}[\log p_\theta(x|w, z)] - E_{q_\phi(w|x)}[\log p_\gamma(y|w)] + D_{KL}(q_\phi(w, z|x)||p(w, z)). \tag{3}$$

Meanwhile, since the objective function in Eq. (3) does not contribute to our assumption that $z$ is independent from $w$ and $y$, and values in $w$ are independent with each other, we decompose the KL-divergence in Eq. (3) and penalize the term:

$$\mathcal{L}_2 = \rho_1 \cdot D_{KL}(q(z, w)||q(z)q(w)) + \rho_2 \cdot D_{KL}(q(w)||\prod_i q(w_i)), \tag{4}$$

where $\rho_1$ and $\rho_2$ are co-efficient hyper-parameters to penalize the two terms. Details of the proof and derivation regarding the overall objective can be refereed in Appendix A.

### 4.1.2 Relating the properties and latent variables

To model the dependence between the correlated properties and the associated latent variables $p(y|w)$ in Eq (3) as well as to capture the correlation among properties, we propose to directly learn the specific relationship between disentangled latent variables in $w$ and properties $y$. The correlations among $y$ are also captured. Specifically, we design a mask pooling layer achieved by a mask matrix $M \in \{0, 1\}^{l \times m}$, where $l$ is the dimension of the latent vector $w$. $M$ captures the way how $w$ relates to $y$, where $M_{i,j} = 1$ denotes that $w_i$ relates to the $j$-th property $y_j$, otherwise there is no relation. In this way, two properties that relate to the same variable in $w$ can be regarded as correlated. The binary elements in $M$ are trained with the Gumbel Softmax function. In implementation, the $L_1$ norm of the mask matrix is also added to the objective to encourage the sparsity of $M$.

Next, given the learned mask matrix $M$, we model the mapping from $w$ to $y$. For properties $y$, we can calculate the corresponding $w'$ that aggregates the values in $w$ that contribute to each property as $w \cdot J^T \odot M$, each column of which corresponds to the related latent variables in $w$ to be aggregated to predict the corresponding $y$. For each property $y_j$ in $y$, we aggregate all the information from its related latent variable set in $w$ into the next-level latent variable $w'_j$ (i.e., the $j$-th variable of $w'$) via an aggregation function $h$:

$$w' = h(w \cdot J^T \odot M; \beta), \tag{5}$$

where $J$ is a vector with all values as one, $\odot$ represents the element-wise multiplication and $\beta$ is the parameter of $h$. Then the property $y$ can be predicted using $w'$ as:

$$y = f(w'; \gamma), \tag{6}$$

where $f$ is the set of prediction functions with $w' = h(w \cdot J^T \odot M; \beta)$ as the input and $\gamma$ are the parameter which will be further explained in the next section. Thus, we have built a one-to-one mapping between $w'$ and $y$. In addition, the correlation of $y_i$ and $y_j$ can be recovered if $M_{\cdot i}^T \cdot M_{\cdot j} \neq 0$.

### 4.1.3 Invertible constraint for multiple-property control

As stated in the problem formulation, our proposed model aims to generate a data point $x$ that retains the original property value requirement for the given properties. The most straightforward way to do this is to model both the mutual dependence between each $y_i$ and its relevant latent variable set $w'_i$. However, this can incur double errors in this two-way mapping, since there exists a complex correlation among properties in $y$ and there are many cases that $M_{\cdot i}^T \cdot M_{\cdot j} \neq 0$. To address it, we propose an invertible function that mathematically ensures the exact recovery of bridging variables $w'$ given a group of desired properties $y$ based on the following deduction.

As in Eq. (6), the set of correlated properties $y = \{y_1, y_2, ..., y_m\}$ are correlated with the set of latent variables $w' = \{w'_1, w'_2, ..., w'_m\}$ in a one-to-one mapping fashion. Thus we assume that $y$ can be sampled from a multivariate Gaussian given $w'$ as follows:

$$p(y|w') = \mathcal{N}(y|f(w'; \gamma), \Sigma); y = (y_1, y_2, ..., y_m), w' = \{w'_1, w'_2, ..., w'_m\}, \Sigma \in \mathbb{R}^{m \times m}$$

$$s.t., f(w'; \gamma)[j] = \bar{f}(w'; \gamma)[j] + w'_j, Lip(\bar{f}(w'; \gamma)[j]) < 1 \quad \text{if} \quad ||W_k||_2 < 1, j = 1, ..., m, \tag{7}$$

where $Lip$ denotes to the $Lipschitz - constant$. Namely, to precisely control the properties $y$, we learn a set of invertible functions $f(w'; \gamma)$ indicated in Eq 6 to model $p_\gamma(y|w')$. $\gamma$ is the set of parameters in Eq 6. The constraint enforces $f(w'; \gamma)[j]$ to be an invertible function to achieve mutual dependence between $y_j$ and $w'_j$ [2]. As a result, we have the third term of the objective function:

$$\mathcal{L}_3 = -\mathbb{E}_{w' \sim p(w')}[\mathcal{N}(y|f(w'; \gamma), \Sigma)] + ||Lip(\bar{f}(w'; \gamma)[j]) - 1||_2 \tag{8}$$

## 4.2 Multi-objective data generation phase

In this section, we introduce how to control the property of the generated data based on the well-trained model. Based on the problem formulation, we aim to generate data that holds the correlated properties $y = \{y_1, y_2, ..., y_m\}$ via $w$, where the properties need to meet a series of value requirements. We approach the goal by firstly mapping properties $y$ back to $w'$ via the invertible function learned in Eq. (7), then optimizing $w$ under a multi-objective optimization framework. Finally, $w$ is combined with $z$ that controls other aspects of data to generate the data via the objective decoder (Figure 2).

To obtain the corresponding $w'$ from properties $y$, conditioning on the fact that (1) $w' = h(w \cdot J^T \odot M; \beta)$ where $h$ is the aggregation function indicated in Eq (5); and (2) $w \sim p(w)$, we optimize the conditional distribution of $p_\gamma(y|w')$ by maximizing the probability that $y = \hat{y}$ as follows:

$$w' = \underset{w'}{\arg\max} \, p_\gamma(y = \hat{y}|w'), \quad w' = \underset{w'}{\arg\max} \log \mathcal{N}(y = \hat{y}|f(w'; \gamma), \Sigma)$$

$$w' = \underset{w'}{\arg\max} - \sum_{j=1}^{m} \sum_{i=1}^{m} (\hat{y}_i - f(w'; \gamma)[i])(\hat{y}_j - f(w'; \gamma)[j])/\sigma_{ij}, \tag{9}$$

where $\sigma_{ij}$ is the element of the $i$-th row and the $j$-th column of $\Sigma$ in Eq. 7. $\hat{y}$ is the set of properties of interest. The above operations are based on the fact that $y$ is a vector that contains continuous variables. Since $\Sigma$ is unknown, we optimize Eq. 9 under the same condition by alternatively solving the following problem based on the Theorem 4.1:

$$w' = \underset{w'}{\arg\max} - \sum_{j=1}^{m} (\hat{y}_j - f(w'; \gamma)[j])^2 \tag{10}$$

**Theorem 4.1.** *Solution of Eq. 10 is also the solution to optimizing Eq. 9.*

The proof of the theorem 4.1 is trivial and can be referred to Appendix B. In this way, we can obtain $w'$ directly from the invertible function $f(w'; \gamma)$ with $\hat{y}$ as the input.

Lastly, given the optimal $w'$, we aim to realize the requirements that are posed on the correlated properties $y$ of the generated data, such as being specific values, lying in a range or reaching the maximum value. To this end, we naturally formalize the process of searching the satisfied $w^*$ as a multi-objective optimization framework. The requirements can be defined as a range of properties in $y$, $c_{i,1} \le y_i \le c_{i,2}$, while it can also be the exact values when $c_{i,1} = c_{i,2}$. These properties can be divided into two groups based on the requirement of the target value: (1) $\mathbf{Y}_v$ represents the set of properties that are required to be a specific value, namely, $y_i = c_{i,1}$; and (2) $\mathbf{Y}_r$ represents the set of properties, the value of which should lie in a range, namely, $c_{i,1} \le y_i \le c_{i,2}$. Then the overall optimization objective is formalized as:

$$w^* \leftarrow \underset{w \sim p(w)}{\arg\max} \bigcup_{y_i \in \mathbf{Y}_v} \{p_\gamma(y_i = c_{i,1}|w')\}$$

$$s.t. \quad w' = h(w; \beta) = f^{(-1)}(y; \gamma), \quad c_{i,1} \le y_i \le c_{i,2}, \forall y_i \in \mathbf{Y}_r, \tag{11}$$

where $\mathbf{Y_v}$ and $\mathbf{Y_r}$ are defined based on different applications. The border of the range can even be set to be infinity or negative infinity. When $c_{i,2}$ is set to be infinity, $y_i$ is maximized in Eq. (11). When $c_{i,1}$ is set to be negative infinity, $y_i$ is minimized in Eq. (11).

The details of overall practical implementation of the aforementioned distributions to model the whole learning and generation process are presented in Appendix C.

## 5 Experiments

### 5.1 Dataset

We evaluate the proposed and comparison models on two molecular datasets and two image datasets: 1) The **Quaternary Ammonium Compound (QAC)** dataset is a real dataset that contains 462 quaternary ammonium compounds processed by the Minbiole Research Lab [1]. An open-source cheminformatics and machine learning library were used to generate a number of properties or features for each of the compounds, in which *molecular weight* and the *logP* value were used as data properties in our experiments; 2) **QM9** dataset is an enumeration of 134,000 stable organic molecules

---

[1]The Minbiole Research Lab: `http://kminbiol.clasit.org/`

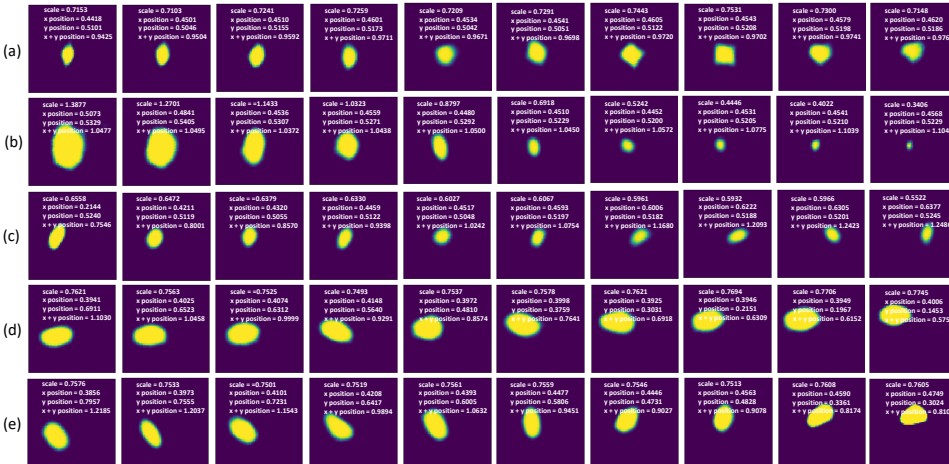

Figure 3: Generated images of CorrVAE by traversing five latent variables in $w$ for dSprites dataset according to the mask matrix (Figure 5). The corresponding properties are illustrated at the top right corner of each image. (a) Traversing on the $w_1$ that only controls *shape*; (b) Traversing on the $w_2$ that only controls *size*; (c) Traversing on the $w_4$ that controls both *x position* and *x+y position*; (d) Traversing on the $w_5$ that controls both *y position* and *x+y position*; (e) Traversing on the $w_3$ that simultaneously controls *x position*, *y position* and *x+y position*.

with up to 9 heavy atoms [39]. *Molecular weight* and the *logP* values serve as the target properties for the comparison with the QAC dataset; 3) **dSprites** contains 737,280 total images regarding 2D shapes procedurally generated from 6 ground truth independent latent factors [33], in which *shape*, *scale*, *x position* and *y position* were employed in our experiments. To construct correlated properties, we additionally formed and tested a new property, *x+y positions* by summing up *x position* with *y position*; and 4) **Pendulum** dataset was originally synthesized to explore causality of the model [46]. Pendulum contains 7,308 images in total with 3 entities (*pendulum, light, shadow*) and 4 properties ((*pendulum angle, light position*) → (*shadow position, shadow length*)).

## 5.2 Comparison models

We compare the proposed model with three comparison models that can be used for property controllable generation: 1) **semi-VAE** pairs the latent space with desired properties and minimizes MSE between latent variables and desired properties during the training process [31]; 2) **CSVAE** correlates the subset of latent variables with properties by minimizing the mutual information [28]; 3) **PCVAE** implements an invertible mapping between each pair of latent variables and desired properties, while it solely relies on the disentanglement assumption of the learned latent space and is short for capturing correlated properties [13]. Besides three comparison models, we consider two other models adapted from CorrVAE for the ablation study: 1) **CorrVAE-1**: we replace the mask pooling layer of the proposed model with the ground-truth mask that is manually obtained from the data; 2) **CorrVAE-2**: We replace the MLP that maps $w$ to $w'$ with the simple linear regression to evaluate the significance of non-linear correlation among properties. The linear regression model employs the $w'$ as the response variable while the corresponding values in $w$ as the independent variables.

## 5.3 Quantitative evaluation

### 5.3.1 Evaluation metrics

In this section, we quantitatively evaluate the proposed model and comparison models on both molecular and image datasets. For image data, we evaluate the model by controlling three properties *shape*, *size*, *x position*, *y position* and *x+y position*.

**Molecule generation evaluation metrics**. We evaluate the performance of molecule generation via three common evaluation metrics which focus on the validity, novelty and uniqueness of the generated molecules as follows: **Validity** measures the percentage of valid molecules overall generated molecules. **Novelty** measures the percentage of new molecules in the generated molecules

that are not in the training set. **Uniqueness** measures the percentage of the unique molecules overall generated molecules. The results are shown in Appendix, Table **??**.

**Controllable molecule generation evaluation metrics**. To evaluate the performance of controllable generation on molecules, we evaluate the proposed model in generating molecules with desired properties. Specifically, we evaluate the MSE between the molecular properties of the generated molecules and the expected molecular properties. The results based on predicting three properties *size*, *x position* and *x+y position* are shown in Table 1.

**Image property prediction evaluation metrics**. As an additional benefit of the proposed model, the prediction of image property from the latent space could be utilized as an image property predictor. On the other side, the prediction performance of the predictor could reflect the quality of the model in learning image properties. We evaluate MSE between the ground truth image property value and the predicted image property value. The results based on predicting three properties *size*, *x position* and *x+y position* are shown in Table 1. We also quantitatively evaluate the quality of generated images via the FID score, reconstruction error and negative log likelihood as shown in Appendix Table **??**.

**Disentanglement evaluation metrics**. We evaluate the disentanglement of latent variables from different perspectives. First of all, we borrow the metric *avgMI* [31] to evaluate the overall performance of the proposed model and comparison models. *avgMI* measures the mutual dependence between latent variables and properties, and is calculated by the Frobenius norm regarding the mutual information matrix and the ground-truth mask matrix: $avgMI = ||I(w,y) - \tilde{M}||_F^2$, where $I(w,y)$ is the pairwise mutual information matrix between latent variables of $w$ and properties. $\tilde{M}$ is the ground-truth mask matrix indicating of the contribution of latent variables to properties. The results are show in Appendix Table **??**. Note that for CorrVAE, instead we calculate $avgMI$ using $I(w',y)$.

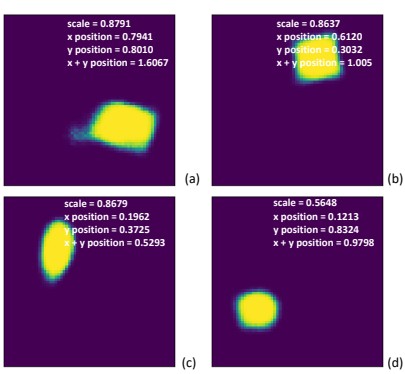

Figure 4: Generation of dSpirates images under different constraints. Properties extracted via pre-trained models are illustrated at the top right corner of each image. (a) *shape*=1 (square), *size*=0.9, *x position*=0.8, *y position*=0.8, *x+y position* = 1.6; (b) *shape*=1 (square), *size*=0.9, *x position*=0.6, *y position*$\in$ $[0.3, 0.4]$; (c) *shape*=2 (ellipse), *size*=0.9, *x position*=$-\infty$, *y position*=0.4; (d) *shape*=2 (ellipse), *size*=0.5, *x position*=$-\infty$, *y position*= $\infty$

### 5.3.2 Overall performance

We use avgMI to evaluate the overall performance of the proposed model and comparison models. As shown in Appendix Table **??**, CorrVAE achieves aligned avgMI with PCVAE and semi-VAE in both dSprites and Pendulum datasets, the value of which is rather small, suggesting that CorrVAE achieves a strong mutual dependence between the latent variables and the corresponding properties. By contrast, CSVAE has a larger avgMI, which results from the fact that CSVAE bonds the property with all latent variables rather than a single variable in $w$. This might affect the efficiency of the model to enforce the mutual dependence between latent variables and properties. We also visualize generated images of CorrVAE in Appendix Figure 1 and they all look similar with those in dSprites. According to Appendix Table **??** and Appendix Table **??**, both molecule and image data are generated well by the proposed model since CorrVAE achieves $100\%$ validity and novelty on molecular generation and comparable reconstruction error and FID values on image generation.

### 5.3.3 Property prediction evaluation

We evaluate the learning ability of the proposed model and comparison models by the MSE between predicted properties and the true properties. As shown in Table 1, CorrVAE achieves the lowest *size* MSE of 0.0016 compared with comparison models, except CSVAE. Nevertheless, CorrVAE has a smaller MSE (0.0066) of *x+y position* than CSVAE (0.3563), showing the superior ability of CorrVAE to handle correlated properties. On Pendulum dataset, CorrVAE achieves the lowest MSE of *pendulum angle* (36.37) among all models except Semi-VAE (9.9455). This might result from the fact that the correlation of properties in Pendulum dataset is much more complex than that in dSprites, which is hard for the mask layer to capture. However, the performance of CorrVAE is also satisfying

Table 1: CorrVAE compared to state-of-the-art methods on dSprites and Pendulum datasets according to MSE between predicted correlated properties and true properties.

| Method | dSprites | | | Pendulum | | | |
|---|---|---|---|---|---|---|---|
| | size | x position | x+y position | pendulum angle | light position | shadow position | shadow length |
| CSVAE | 0.0006 | 0.0005 | 0.3563 | 184.4047 | 108.9864 | 29.6706 | 4.2027 |
| Semi-VAE | 0.0031 | 0.0030 | 0.0031 | 9.9455 | 6.5635 | 1.3296 | 0.7315 |
| PCVAE | 0.0038 | 0.0037 | 0.0038 | 37.3644 | 12.2372 | 2.2977 | 0.7669 |
| CorrVAE-1 | 0.0024 | 0.0059 | 0.0023 | 39.9255 | 11.2878 | 6.3579 | 2.4626 |
| CorrVAE-2 | 0.0019 | 0.0098 | 0.0088 | 1555.33 | 475.6341 | 360.1743 | 72.5489 |
| CorrVAE | 0.0016 | 0.0077 | 0.0066 | 36.3700 | 15.3900 | 6.0250 | 10.2600 |

regarding the MSE of *light position* and *shadow position*, compared with comparison models. Not surprisingly, as shown in Table 1, CorrVAE-1 achieves better performance than CorrVAE on those correlated properties since it uses the ground-truth mask to control the correlation among properties, which is challenging for CorrVAE to learn. Besides, the performance of CorrVAE-1 and CorrVAE are rather close on independent properties such as *size* for dSprites and *pendulum angle* as well as *shadow position* for Pendulum. This might be due to that *size* is independent with two other variables in dSprites while *pendulum angle* and *shadow position* are independent conditioning on *light position* and *shadow position* in Pendulum. The independence can be well captured by CorrVAE (Figure 5 and Appendix Figure 4) so that it has comparable performance on those variables with CorrVAE-1. CorrVAE-2 has worse performance in both dSprites and Pendulum datasets, as shown in Table 1. This is because that CorrVAE-2 models $\tilde{w}$ to $w'$ using simple linear regression, which cannot capture the non-linear correlation among properties that might exist in the dataset.

### 5.3.4 Controllable generation quality

As shown in Appendix Table **??**, for the MSE between generated and expected MolWeight compared with comparison models, CorrVAE achieves the lowest MSE of MolWeight(356701.5) and the MSE of logP(24.01) only larger than the Semi-VAE(15.13) on the QAC dataset; CorrVAE achieves a comparable MSE(logP: 2.75, MolWeight: 4476.54) on the QM9 dataset. Meanwhile, CorrVAE has a lower MSE between generated and expected logP than Semi-VAE. We also evaluate the quality of generated molecules based on the QAC and QM9 dataset, in which CorrVAE and all comparison models have a satisfying performance (Appendix Table **??**).

### 5.4 Qualitative evaluation

We qualitatively evaluate the ability of the property control of CorrVAE using the dSprites dataset. We firstly interpret the mask pooling layer learned in the training process. Then we visualize the change of properties when manipulating corresponding latent variables in $w$. Lastly, we visualize the images that preserve target properties achieved based on section 4.2. In addition, we pre-trained four predictors to extract properties, including *shape*, *scale*, *x position*, *y position* and *x+y position*, from generated images. All these models were pre-trained in the training set of CorrVAE with the MSE less than 0.001. The generated images in Figure 3 and Figure 4 are annotated by their corresponding features extracted by these pre-tained models. Details regarding the structure of pre-trained models are explained in Appendix Table **??**.

### 5.4.1 Interpreting mask matrix

We train CorrVAE using *shape*, *scale* and three correlated properties *x position*, *y position* and *x+y position* while setting the dimension of $w$ as 8. As shown in Figure 5, eventually we obtained an interpretable mask matrix indicating that $w_1$ only controls *shape* and $w_2$ only controls *scale*. This is aligned with the fact that those two properties are independent with others in the data. We also observed that $w_3$ simultaneously controls *x position*, *y position* and *x+y position*, indicating that they are correlated since *x+y position* is generated from *x position* and *y position*. The correlation among those three variables is also captured by $w_4$ that simultaneously controls *x position* and *x+y position*, and $w_5$ that simultaneously controls *y position* and *x+y position*. In addition, there is one single variable $w_6$ that only controls *y position* and another single variable $w_8$ that only controls *x position*.

### 5.4.2 Property control by manipulating latent variables

The mask matrix learned in the training process (Figure 5) enables the mask pooling layer of CorrVAE (Figure 2) to control properties accordingly. Based on the mask matrix shown in Figure 5, as shown in Figure 3 (a), we traverse the value of $w_1$ within $[-5, 5]$ and the *shape* of the pattern changes accordingly from ellipse to square. Moreover, we traverse the value of $w_2$ that controls

*size* of the object within $[-5, 5]$ and expectedly, the *size* of the pattern keep decreasing from 1.3877 to 0.3406, as shown in Figure 3 (b). If we traverse on $w_4$ that controls both *x position* and *x+y position* within $[-5, 5]$, we find that the *x position* of the object moves from the left to the right while *x+y position* changes accordingly (Figure 3 (c)). Similarly, when we traverse $w_5$ that controls both *y position* and *x+y position* within $[-5, 5]$, the *y position* moves from the bottom to the top (Figure 3 (d)) while *x+y position* also changes accordingly. We also evaluate the more complex setting by traversing the value of $w_3$ within $[-5, 5]$ that simultaneously controls *x position*, *y position* and *x+y position*. Not surprisingly, the position of the pattern changes in both horizontal and vertical directions, corresponding to *x+y position*. At the mean time, *x position* and *y position* change accordingly, as shown in Figure 3 (e). We also showcase the whole batch (eight) of generated images in Appendix Figure 2 corresponding to each constraint of Figure 4. All images for the same constraint look similar, indicating the consistency and the replicability of our model.

### 5.4.3 Multi-objective generation

We evaluate the performance of CorrVAE for the multi-objective generation on the dSprites dataset. The experiments are performed based on the model that controls five properties, *shape*, *scale*, *x position*, *y position* and *x+y position*, and the mask matrix learned from the training process (Figure 5). Four sets of property constraints are considered and the corresponding $w^*$'s are obtained according to section 4.2. Then the image is generated by the object decoder of CorrVAE from $w^*$. Specifically, when the set of target properties is *shape*=square, *size*=0.9, *x position*=0.8, *y position*=0.8 and *x+y position*=1.6, the location of the pattern leans roughly toward the lower right corner and has a shape of square (Figure 4 (a)). Besides, when the constraints of target properties are set as *shape*=square, *size*=0.9, *x position*=0.6 and *y position*$\in [0.3, 0.4]$, we can visualize that the generated pattern has a large size and is located towards the right hand side, as shown in Figure 4 (b). If we minimize the *x*

| | $w_1$ | $w_2$ | $w_3$ | $w_4$ | $w_5$ | $w_6$ | $w_7$ | $w_8$ |
|---|---|---|---|---|---|---|---|---|
| shape | 1 | 0 | 0 | 0 | 0 | 0 | 0 | 0 |
| size | 0 | 1 | 0 | 0 | 0 | 0 | 0 | 0 |
| x position | 0 | 0 | 1 | 1 | 0 | 0 | 0 | 1 |
| y position | 0 | 0 | 1 | 0 | 1 | 1 | 0 | 0 |
| x+y position | 0 | 0 | 1 | 1 | 1 | 0 | 0 | 0 |

Figure 5: The mask matrix learned by the training process. Each column corresponds to one latent variable in $w$. Each row corresponds to a property. In our experiments setting, five properties, *shape*, *scale*, *x position*, *y position* and *x+y position*, are handled.

*position* and set the *shape*=ellipse, *size*=0.9 and *y position*=0.4, as shown in Figure 4 (c), a pattern of an ellipse with large size but at the very top of the image is observed while the *y position* roughlt aligns with the constraint. Lastly, we decrease the size, minimize *x position* and maximize *y position*. Based on Figure 4 (d), the generated pattern is located at the very lower left corner and has a much smaller size compared with other generated images. In conclusion, the results show our model can generate objects with target properties based on the multi-objective optimization framework. All properties including shape are roughly aligned with the constraints, as shown in Figure 4.

## 6 Conclusion

In this paper, we attempt to tackle several challenges in multi-objective data generation by proposing a novel deep generative model. Firstly, we identify three challenges in generating data preserving target correlated properties. Secondly, we propose CorrVAE that includes a mask pooling layer to identify and control correlation among properties, and a multi-objective optimization framework to generate data with desired properties. Comprehensive experiments were conducted on real-world datasets and our model shows superior performance than comparison models. Future work will be done on testing other multi-objective optimiation techniques.

### Acknowledgments and Disclosure of Funding

This work was supported by the NSF Grant No. 2007716, No. 2007976, No. 1942594, No. 1907805, No. 1841520, No. 1755850, Meta Research Award, NEC Lab, Amazon Research Award, NVIDIA GPU Grant, and Design Knowledge Company (subcontract number: 10827.002.120.04).

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
