# A Derivation of the overall objective

The goal requires us to not only model the dependence between $x$ and $(w, z)$ for latent representation learning and data generation, but also model the dependence between $y$ and $w$ for controlling the property. We propose to achieve this by maximizing the joint log likelihood $p(x, y)$ via its variational lower bound. Given an approximate posterior $q(z, w|x, y)$, we can use the Jensen's equality to obtain the variational lower bound of $\log p(x, y)$ as:

$$
\begin{aligned}
\log p(x, y) &= \log \mathbb{E}_{q(z,w|x,y)}[p(x, y, w, z)/q(z, w|x, y)] \\
&\geq \mathbb{E}_{q(z,w|x,y)}[\log p(x, y, w, z)/q(z, w|x, y)].
\end{aligned} \tag{1}
$$

The joint likelihood $\log p(x, y, w, z)$ can be further decomposed as $\log p(x, y|z, w) + \log p(z, w)$. Two assumptions apply to our task: (1) $x$ and $y$ are conditionally independent given $w$ (i.e., $x \perp y|w$) since $w$ only captures information from $y$; (2) $z$ is independent from $w$ and $y$, equivalent to $y \perp z|w$. This gives us $x \perp y|(w, z)$, suggesting that $\log p(x, y|w, z) = \log p(x|w, z) + \log p(y|w, z) = \log p(x|w, z) + \log p(y|w)$.

*Proof.* Proof of $x \perp y|(w, z)$ given $x \perp y|w, y \perp z, y \perp w$.

Firstly we will prove that given $z \perp y$ and $z \perp w$, we have $y \perp z|w$. Based on the Bayesian rule, we have:

$$
p(y, z|w) = p(z|y, w)p(y|w) = p(y|z, w)p(z|w) \tag{2}
$$

Since $z \perp y$ and $z \perp w$, then we have $p(z|w) = p(z)$ and $p(z|w, y) = p(z)$. As a result, both sides of Eq. 2 can be cancelled as $p(z)p(y|w) = p(y|z, w)p(z)$, causing $p(y|w) = p(y|z, w)$. We multiply $p(z|w)$ then we have $p(z|w)p(p(y|w) = p(y, z|w)$. Thus, we have $y \perp z|w$

Then, given that $x \perp y|w, y \perp z, y \perp w$ and $y \perp z|w$, based on the Bayesian rule, we have $p(x, y|w, z) = p(y|x, z, w)p(x|z, w) = p(x|y, z, w)p(y|z, w)$. This equation can be cancelled as $p(y|w)p(x|z, w) = p(x|y, z, w)p(y|w)$ given $y \perp z|w$ and $y \perp x|w$. Then we have $p(x|z, w) = p(x|y, z, w)$, indicating that $x \perp y|(w, z)$. $\qquad\square$

Consequently, we can write the joint log-likelihood and maximize its lower bound as:

$$
\begin{aligned}
\log p_{\theta,\gamma}(x, y, w, z) &= \log p_\theta(x|w, z) + \log p(w, z) + \log p_\gamma(y|w) \\
&= \log p_\theta(x|w, z) + \log p(w, z) + \sum_{i=1}^{m} \log p_\gamma(y_i|w_i'),
\end{aligned} \tag{3}
$$

where we define $w_i'$ as the set of values in $w$ that contribute to the $i-th$ property to bridge the mapping $w \to y$ and allow property controlling.

*Proof.* Proof of $\log p_\gamma(y|w) = \sum_{i=1}^{m} \log p_\gamma(y_i|w_i')$.

Since $w'$ aggregates all information in $w$, we have $\log p_\gamma(y|w) = \log p_\gamma(y|w')$, Also, since properties in $y$ are independent conditioning on $w'$, $\log p_\gamma(y|w) = p_\gamma(y|w') = \sum_{i=1}^{m} \log p_\gamma(y_i|w') = \sum_{i=1}^{m} \log p_\gamma(y_i|w_i')$. $\qquad\square$

Given $q_\phi(w, z|x, y) = q_\phi(w, z|x) = q_\phi(w|x) \cdot q_\phi(z|x)$ since the information of $y$ is included in $x$, we rewrite the aforementioned joint probability as the form of the Bayesian variational inference:

$$
\begin{aligned}
\mathcal{L}_1 = -&\mathbb{E}_{q_\phi(w,z|x)}[\log p_\theta(x|w, z)] - E_{q_\phi(w|x)}[\log p_\gamma(y|w)] \\
&+ D_{KL}(q_\phi(w, z|x)||p(w, z)).
\end{aligned} \tag{4}
$$

Since the objective function in Eq. (3) does not achieve our assumption that $z$ is independent from $w$ and $y$, we decompose the KL-divergence in Eq. (4) as:

$$
\begin{aligned}
\mathbb{E}_{p(x)}[D_{KL}(q_\phi(w, z|x)||p(w, z))] = &D_{KL}(q_\phi(w, z, x)||q(w, z)p(x)) \\
&+ D_{KL}(q(w, z)||\prod_{i,j} q(z_i)q(w_j)) \\
+ \sum_i D_{KL}(q(z_i)||p(z_i)) &+ \sum_j D_{KL}(q(w_j)||p(w_j)),
\end{aligned} \tag{5}
$$

where $z_i$ is the $i$-th variable of the latent vector $z$ and $w_j$ is the $j$-th variable of the latent vector $w$. Then we can further decompose the total correlation term in Eq. (5) as:

$$D_{KL}(q(w,z)||\prod_{i,j} q(z_i)q(w_j)) = D_{KL}(q(z,w)||q(z)q(w))$$

$$+D_{KL}(q(w)||\prod_i q(w_i)) + D_{KL}(q(z)||\prod_i q(z_i)) \qquad (6)$$

Thus, we can add a penalty of the first term of Eq. 6 to enforce the independence between $z$ and $w$ and another penalty to the second term to enforce the independence of variables in $w$. This is the second term of our final objective with the hyper-parameter $\rho$ to penalize the term:

$$\mathcal{L}_2 = \rho_1 \cdot D_{KL}(q(z,w)||q(z)q(w)) + \rho_2 \cdot D_{KL}(q(w)||\prod_i q(w_i)), \qquad (7)$$

$\mathcal{L}_1 + \mathcal{L}_2$ is the overall objective of our model. Together with the third term as illustrated in the main text:

$$\mathcal{L}_3 = -\mathbb{E}_{w' \sim p(w')}[\mathcal{N}(y|f(w';\gamma), \Sigma)] + ||Lip(\bar{f}(w';\gamma)[j]) - 1||_2 \qquad (8)$$

Our final objective is $\mathcal{L}_1 + \mathcal{L}_2 + \mathcal{L}_3$.

# B  Proof of Theorem 4.1

*Proof.* We will prove Theorem 4.1 by taking the derivative of the objective function in both Eq. (9) and Eq .(10) regarding $w'$. Suppose $g_1(w')$ and $g_2(w')$ are objective function of Eq. (9) and Eq. (10), respectively. To simplify the proof, rewrite $g_1(w')$ and $g_2(w')$ in the matrix form as:

$$g_1(w') = -(\hat{y} - f(w';\gamma))^T \Sigma^{-1}(\hat{y} - f(w';\gamma))$$

$$g_2(w') = -(\hat{y} - f(w';\gamma))^T(\hat{y} - f(w';\gamma))$$

Then we take the derivative of $g_1(w')$ on $w'$:

$$\frac{\partial g_1(w')}{\partial w'} = \frac{\partial g_1(w')}{\partial f(w';\gamma)} \frac{\partial f(w';\gamma)}{\partial w'} = 0$$

Since $f(w';\gamma)$ is the prediction function, it is not necessary for $f(w';\gamma)$ to reach maximum or minimum value at $w'$ all the time. And the above equation can always been satisfied if $\frac{\partial g_1(w')}{\partial f(w';\gamma)} = 0$. Then we have:

$$\frac{\partial g_1(w')}{\partial f(w';\gamma)} = 0$$
$$\rightarrow 2(\hat{y} - f(w';\gamma))^T \Sigma^{-1} = 0$$
$$\rightarrow (\hat{y} - f(w';\gamma))^T = 0$$
$$\hat{y}_i = f(w';\gamma)[i], i = 1,...,m, \qquad (9)$$

assuming $\Sigma$ is positive definite. Similarly for $g_2(w')$, we have:

$$\frac{\partial g_2(w')}{\partial f(w';\gamma)} = 2(\hat{y} - f(w';\gamma)) = 0$$
$$\hat{y}_i = f(w';\gamma)[i], i = 1,...,m \qquad (10)$$

Thus, Eq 9 and Eq 10 share the same set of solution, suggesting that the solution to Eq. (10) is also a solution to Eq. (9). □

# C  The Overall Implementation

In this section, we introduce the overall implementation of the aforementioned distributions to model the whole learning and generation process. All experiments are conducted on the 64-bit machine with a NVIDIA GPU, NVIDIA GeForce RTX 3090.

Table 1: Implementation details of CorrVAE on image data (dSprites and Pendulum). Conv represents the layer of convolutional neural network; ConvTranspose represents the transposed convolutional layer; ReLU represents the Rectified Linear Unit activation function; FC is the fully connected layer.

| Layer | Object encoder | Property encoder | Object decoder |
|---|---|---|---|
| Input | $x$ (image) | $x$ (image) | $z$ and $w$ |
| Layer1 | Conv+ReLU | Conv+ReLU | FC+ReLU |
| Layer2 | Conv+ReLU | Conv+ReLU | FC+ReLU |
| Layer3 | Conv+ReLU | Conv+ReLU | FC+ReLU |
| Layer4 | Conv+ReLU | Conv+ReLU | ConvTranspose+ReLU |
| Layer5 | FC+ReLU | FC+ReLU | ConvTranspose+ReLU |
| Layer6 | FC+ReLU | FC+ReLU | ConvTranspose+ReLU |
| Layer7 | FC | FC | ConvTranspose+ReLU |
| Output | $z$ | $w$ | $x$ (image) |

Table 2: Implementation details of CorrVAE on molecular data (QAC and QM9). GGNN represents the gated graph neural network; ReLU represents the Rectified Linear Unit activation function; FC is the fully connected layer.

| Layer | Object encoder | Property encoder | Object decoder |
|---|---|---|---|
| Input | $G$ (molecule) | $G$ (molecule) | $z$ and $w$ |
| Layer1 | FC+ReLU | FC+ReLU | FC+ReLU |
| Layer2 | GGNN+ReLU | GGNN+ReLU | GGNN+ReLU |
| Layer3 | GGNN+ReLU | GGNN+ReLU | GGNN+ReLU |
| Layer4 | FC | FC | FC (for both node and edge) |
| Output | $z$ | $w$ | $G$ (molecule) |

We have two encoders to model the distribution $q(w, z|x)$, and two decoders to model $p(y|w)$ and $p(x|w, z)$ for property control and data generation, respectively. For the first objective $\mathcal{L}_1$ (Eq. (3)), we use Multi-layer perceptrons (MLPs) together with Convolution Neural Networks (CNNs) or Graph Neural Networks (GNNs) for image or graph data, respectively, to capture the distribution over relevant random variables. For $\mathcal{L}_2$ in Eq. (4), since both $q(z)$ and $q(w)$ are intractable, we use Naive Monte Carlo approximation based on a mini-batch of samples to approach $q(z)$ and $q(w)$ [2]. The details regarding the architecture of CorrVAE on image and molecular datasets are presented in Table 1 and Table 2, respectively. The dimension of each layer can be tuned based on different needs.

The mask layer $M$ is formed and trained with the Gumbel Softmax function, while $h$ function in Eq. (5) is modeled by MLPs. The $L_1$ norm of the mask matrix is added to the objective to encourage the sparsity of the mask matrix. The invertible constraint and modeling $p_\gamma(y|w')$ in Eq. (7) are achieved by MLPs, by which $\bar{f}(w'; \gamma)[j]$ is approximated with $j = 1, 2, ..., m$, and $f(w'; \gamma)[j] = \bar{f}(w'; \gamma)[j] + w'_j$, as in the constraint of Eq. (7). Since the function $\bar{f}(w'; \gamma)[j]$ approximated by MLPs contains operation of nonlinearities (e.g., ReLU, tanh) and linear mappings, then we have $Lip(\bar{f}(w'; \gamma)[j]) < 1$ if $||W_l||_2 < 1$ for $l \in L$, where $W_l$ is the weights of the $l$-th layer in $\bar{f}(w'; \gamma)[j]$. $|| \cdot ||$ is the spectral norm and $L$ is the number of layers in MLPs. To apply the above constraints, we use the spectral normalization for each layer of MLPs [1].

For generating data with desired properties, we borrow the weighted-sum strategy to solve the multi-objective optimization problem in Eq. (11) to obtain the corresponding $w^*$. We formalize the inequality constraint in Eq. (11) into the KKT conditions. Then $w^*$ serves as the input to the generator of the trained model to generate objects with desired properties.

The pre-trained models to predict properties given an image are trained on all data from dSprites dataset, and the structure of pre-trained models is as below (Table 3):

# D   Quantitative evaluation

The quality of generated molecules based on QAC and QM9 datasets is evaluated by *validity*, *novelty* and *uniqueness*. The results have been shown in Table 4. The quality of generated images is evaluated by *negative log probability* ($-\log$Prob) and *FID* as shown in Figure 5.

Table 3: Implementation details of pre-trained models on dSprites dataset to predict properties $y$. Conv represents the layer of convolutional neural network; ConvTranspose represents the transposed convolutional layer; ReLU represents the Rectified Linear Unit activation function; FC is the fully connected layer.

| Layer | Model |
|---|---|
| Input | Conv+ReLU |
| Layer1 | Conv+ReLU |
| Layer2 | Conv+ReLU |
| Layer3 | FC+ReLU |
| Layer4 | FC |
| Output | $y$ |

Table 4: Generation quality of each method regarding validity, novelty and uniqueness on QAC dataset.

| 2*Method | QAC | | | QM9 | | |
|---|---|---|---|---|---|---|
| | validity | novelty | uniqueness | validity | novelty | uniqueness |
| Semi-VAE | 100% | 100% | 37.5% | 100% | 100% | 82.5% |
| PCVAE | 100% | 100% | 89.2% | 100% | 99.6% | 92.2% |
| CorrVAE | 100% | 100% | 44.5% | 100% | 91.2% | 23.8% |

Table 5: Generation quality of each method regarding validity, novelty and uniqueness on dSprites dataset.

| Method | $-\log\text{Prob}$ | Rec. Error | FID |
|---|---|---|---|
| CSVAE | 0.26 | 227 | 86.14 |
| Semi-VAE | 0.23 | 239 | 86.05 |
| PCVAE | 0.23 | 222 | 85.45 |
| CorrVAE | 0.22 | 229 | 85.17 |

Table 6: The avgMI achieved by each model on the dSprites and Pendulum datasets.

| Method | dSprites | Pendulum |
|---|---|---|
| CSVAE | 0.1578 | 0.1099 |
| Semi-VAE | 0.0118 | 0.0223 |
| PCVAE | 0.0119 | 0.0252 |
| CorrVAE | 0.0404 | 0.0468 |

Table 7: CorrVAE compared to state-of-the-art methods on QAC datasets according to MSE between generated correlated properties and expected properties.

| 2*Method | QAC | | QM9 | |
|---|---|---|---|---|
| | logP | MolWeight | logP | MolWeight |
| Semi-VAE | 15.13 | 433447.6 | 50.55 | 47365.07 |
| PCVAE | 29.76 | 365098.7 | 2.33 | 4528.7 |
| CorrVAE | 24.01 | 356701.5 | 2.75 | 4476.54 |

Table 8: CorrVAE compared to Bayesian optimization on dSprites and Pendulum datasets according to MSE between predicted correlated properties and true properties.

| 2*Method | dSprites | | Pendulum | |
|---|---|---|---|---|
| | size | x+y position | light position | shadow position |
| BO | 0.0033 | 0.0062 | 19.2387 | 17.2858 |
| CorrVAE | 0.0016 | 0.0066 | 15.3900 | 6.0250 |

We also conducted additional experiments by predicting properties with the whole $w$ and performing property control via Bayesian optimization. In this case $w'$ and the mask layer are dropped. The results that compare CorrVAE and the Bayesian optimization-based model (BO) are shown in Table 8. Based on the results, CorrVAE achieves smaller MSE on both light position and shadow position of Pendulum dataset. Specifically, for light position, MSE achieved by CorrVAE is 15.3900 which is much smaller than 19.2387 obtained from the Bayesian optimization-based model. For shadow position, MSE achieved by CorrVAE is 6.0250 which is much smaller than 17.2858 obtained from the Bayesian optimization-based model. Besides, on dSprites dataset, CorrVAE achieves the MSE of 0.0016 for the size which is smaller than 0.0033 obtained from the Bayesian optimization-based model. In addition, CorrVAE achieves comparable results on x+y position with the Bayesian optimization-based model. The results indicate that CorrVAE has better performance than the

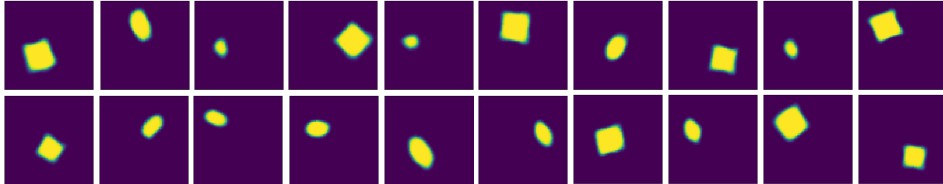

Figure 1: Visualize generated images from CorrVAE.

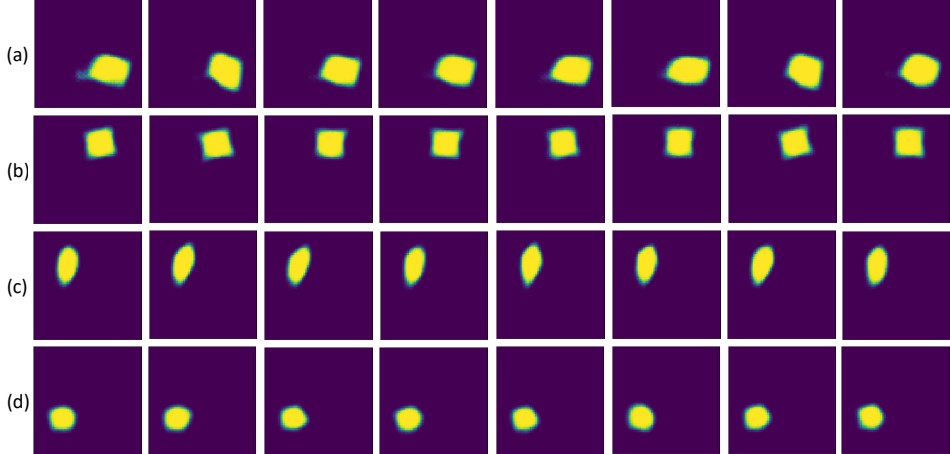

Figure 2: Show case of eight generated images in a batch corresponding to Figure 4 in the main text. (a) *shape*=1 (square), *size*=0.9, *x position*=0.8, *y position*=0.8, *x+y position* = 1.6; (b) *shape*=1 (square), *size*=0.9, *x position*=0.6, *y position*$\in [0.3, 0.4]$; (c) *shape*=2 (ellipse), *size*=0.9, *x position*=$-\infty$, *y position*=0.4; (d) *shape*=2 (ellipse), *size*=0.5, *x position*=$-\infty$, *y position*= $\infty$

Bayesian optimization-based model on controlling independent variables (i.e., size in dSprites, light position in Pendulum) and correlated properties (shadow position in Pendulum).

# E  Qualitative evaluation

We evaluate the property controllability of our model by traversing the latent variables in $w$ that control corresponding properties. In addition to controlling all five properties of the dSprites dataset, we also conducted a naive experiment to control three properties *size*, *x position* and *x+y position* (Figure 4 and Figure 3). Figure 4 shows that mask matrix learned by the model indicating latent variables that control corresponding properties. Specifically, we argue that two variables, $w_6$ and $w_8$ can only control *y position* and *x position*, respectively, as indicated by the mask matrix learned from the training process. As shown in Figure 3 and Figure 5, if we traverse $w_3$ that only controls *x position* (Appendix Figure 4), the horizontal position of the object will move from the left to the right (Appendix Figure 3 (a)) while *x+y position* keeps unchanged but *y position* cannot be controlled since its information is not captured by $w$ and the mask matrix.

In addition to traversing latent variables, we also performed multi-objective optimization on images according to different constraints of properties (Figure 5). Since we do not control *shape* of those images so that this property can go random in the generation process, while all other properties are well controlled by the multi-objective optimization framework.

Moreover, we also traverse the latent variables in $w'$ by simultaneously traversing on latent variables in $w$ corresponding to the associated $w'$ and visualize how the relevant property changes in Figure 6. As is shown in Figure 6 (a), the shape of the pattern changes from ellipse to square as we traverse on $w'_1$. In Figure 6 (b), the size of the pattern shrinks as we traverse on $w'_2$. In Figure 6 (c), the *x position* of the pattern moves from left to right as we traverse on $w'_3$. In Figure 6 (d), the *y position* of the

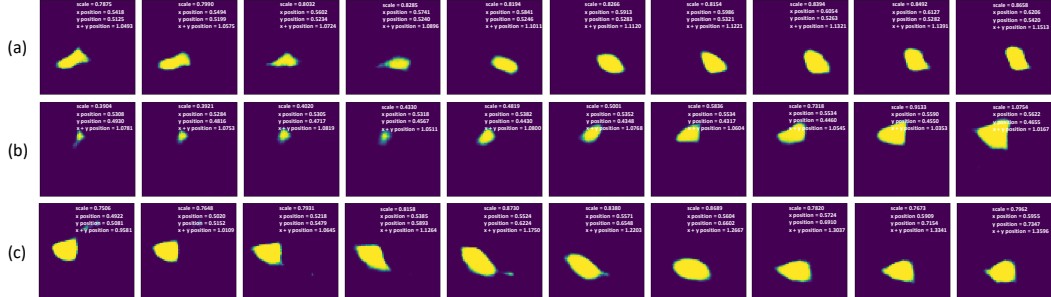

Figure 3: Generated images of corrVAE by traversing three latent variables in $w$ for dSprites dataset according to the mask matrix (Figure 4). The corresponding properties are illustrated at the top right corner of each image. (a) Traversing on the $w_3$ that only controls *x position*; (b) Traversing on the $w_5$ that only controls *size* of the object; (c) Traversing on the $w_7$ that simultaneously controls both *x position* and *x+y position*

|  | $w_1$ | $w_2$ | $w_3$ | $w_4$ | $w_5$ | $w_6$ | $w_7$ | $w_8$ |
|---|---|---|---|---|---|---|---|---|
| *scale* | 0 | 0 | 0 | 0 | 1 | 0 | 0 | 0 |
| *x position* | 0 | 0 | 1 | 0 | 0 | 0 | 1 | 0 |
| *x+y position* | 1 | 0 | 0 | 0 | 0 | 0 | 1 | 0 |

Figure 4: The mask matrix learned by the training process. Each column corresponds to one latent variable in $w$. Each row corresponds to a property. In our experiments setting, three properties, *scalre*, *x position* and *x+y position*, are handled. *x position* and *x+y position* are correlated properties

pattern moves from top to bottom as we traverse on $w_4'$. In Figure 6 (e), the *x position*, *y position* and *x+y position* of the pattern simultaneously change as we traverse on $w_4'$, where *x position* moves from left to right, *y position* moves from bottom to top and *x+y position* increase.

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

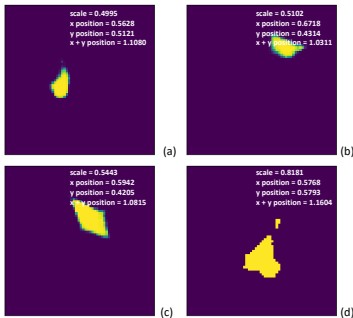

Figure 5: Generation of dSpirates images under different constraints. Properties extracted via pretrained models are illustrated at the top right corner of each image. (a) *scale*=0.5, *x position*=0.5, *x+y position* = 1; (b) *scale*=0.5, *x position* $\in (0.7, 0.9)$, *x+y position*=1; (c) *scale*=0.5, *x position*=0.6, *x+y position*=1; (d) *scale*=0.8, *x position*=0.5, *x+y position*= $\infty$

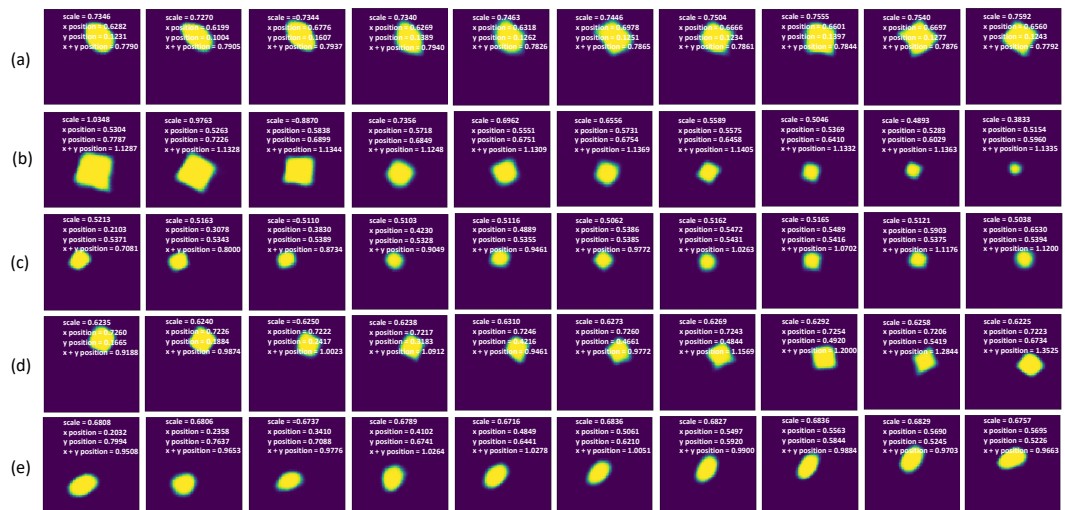

Figure 6: Generated images of corrVAE by traversing five latent variables in $w'$ for dSprites dataset according to the mask matrix (Figure 5). The corresponding properties are illustrated at the top right corner of each image. (a) Traversing on the $w'_1$ that controls *shape*; (b) Traversing on the $w'_2$ that controls *size*; (c) Traversing on the $w'_3$ that controls *x position*; (d) Traversing on the $w'_4$ that controls *y position*; (e) Traversing on the $w'_5$ that controls *x+y position*.

124    man, N. Cesa-Bianchi, and R. Garnett, editors, *Advances in Neural Information Processing*
125    *Systems*, volume 31, 2018. URL https://proceedings.neurips.cc/paper/2018/file/
126    1ee3dfcd8a0645a25a35977997223d22-Paper.pdf.