# OpenReview forum: "Multi-objective Deep Data Generation with Correlated Property Control"
_NeurIPS.cc/2022/Conference — NeurIPS 2022 Accept_

### Official Review · Reviewer_P6bk · 2022-07-08

**Rating:** 5
**Confidence:** 2
**Soundness:** 2 fair
**Presentation:** 2 fair
**Contribution:** 2 fair

**Summary:**

The paper tries to address conditional generation where conditioning properties are correlated.
They point out that existing works that tackle the controllable property generation do not consider the correlation between multiple properties, as it is difficult to identify property correlation. Therefore, they propose a deep generative model that uses invertible mapping to map correlated properties to latent independent variables. During learning, they learn to encode the property and other information through two different encoders and use a novel mask pooling layer on the latent representation corresponding to the property encoder to learn the correlated properties.




**Questions:**



The proposed model seems really interesting but I had some confusion:

1. corresponding properties in figure3 and 4 are not readable.
2. It is not very clear to me how  this conditional property is simplified: q(w,z|x,y) = q(w,z|x)
3. I am not sure why we need the J matrix in w'=wJ^TM , is not the masking matrix M learn if an element of w contribute to w' or not and how big the contribution is?
4. I did not really understand assumption 2 where they derive x and y are independent given z and w.
5. In equation (2), from the first line to the next where they assume y_i can be learned from w'_i, where w_is are independent, is this assumption always holds? so each w'_i is independent, each y_i is derived from one w'_i, but y_i are correlated. So I am a bit confused that if each y_i is only derived from one w'_i, and w'_is are independent, how can they produce a set of y_i that are correlated? This leads to my next question the learned latent w itself stated that elements in w are disentangled from each other, so what is the intension behind why we introduce another set w' that are also disentangled.


**Limitations:**

The proposed idea is very interesting, however, for property-controlled generation, each time it needs to solve the optimization problem to find the proper w for the target property, this could be expensive, also the paper did not compare with the approaches that do bayesian optimization in the latent to find a best patent variable that can generate the data with the target properties.

**Strengths And Weaknesses:**

strength: The problem they try to solve is beneficial for the community and the proposed method is interesting

weakness:  the presentation of the paper could be improved, some parts are bit confusing. They did not compare with the model where they explicitly model the dependency between the properties using hirerichical model or with those who do unconditional generative model coupled with Bayesian optimization to do conditional generation.

---

> ### Author Response · Authors · 2022-08-02
> **Response to Reviewer3 (P6bk) (1/2)**
>
> We appreciate the reviewer's valuable comments and time in helping us improve the manuscript. We reply to each of the points raised below:
>
> **Comment #1**: Corresponding properties in figure3 and 4 are not readable.
>
> **Response #1**: Thank you for the valuable suggestion. We have increased the font size of the annotation in Figure 3 and Figure 4 in the revised paper.
>
> **Comment #2**: It is not very clear to me how this conditional property is simplified: $q(w,z|x,y) = q(w,z|x)$
>
> **Response #2**: This equation is satisfied given the assumption that the information of the y (data properties) is included in $x$ (data) and $y$ can be extracted from $x$, for example,  $y=f(x)$. As a result, $q(w,z|x,y)=q(w, z|x, f(x))=q(w,z|x)$. We have clarified this in Appendix Section A line 29
>
> **Comment #3**: I am not sure why we need the J matrix in $w'=wJ^TM$ , is not the masking matrix $M$ learn if an element of w contribute to $w'$ or not and how big the contribution is?
>
> **Response #3**: Thank you for the opportunity for us to clarify. “$w’=$” is a typo here and has been removed from our original paper (line 175). We use $wJ^TM$ to generate the matrix that formalizes how values in w can be aggregated to $w’$ via the masking matrix $M$ and the aggregation function $h()$ (Eq. 5). Each column of the resulting matrix corresponds to the related latent variables in w to be aggregated to predict the corresponding $y$. The vector $J$ is a vector with all values as one thus does not need to be learned. We have added the clarification in line 178-183.
>
> **Comment #4**: I did not really understand assumption 2 where they derive x and y are independent given z and w.
>
> **Response #4**: Thank you for the question and giving us the opportunity to clarify this. Given $x\perp y\vert w$, $y\perp z, z\perp w$, we have $y\perp z\vert w$ and $x\perp y\vert (w, z)$. Based on the Bayesian rule, we have $p(x, y\vert w, z) = p(y\vert x, z ,w)p(x\vert z, w) = p(x\vert y, z, w)p(y\vert z, w)$. This equation can be canceled as $p(y\vert w)p(x\vert z, w) = p(x\vert y, z, w)p(y\vert w)$ given $y\perp z\vert w$ and $y\perp x\vert w$. Then we have $p(x\vert z, w)=p(x\vert y, z, w)$, indicating that $x\perp y\vert (w, z)$. We have also added the proof to Appendix line12 - 21.
>
> **Comment #5**: In equation (2), from the first line to the next where they assume y_i can be learned from w'_i, where w_is are independent, is this assumption always holds? so each w'_i is independent, each y_i is derived from one w'_i, but y_i are correlated. So I am a bit confused that if each y_i is only derived from one w'_i, and w'_is are independent, how can they produce a set of y_i that are correlated? This leads to my next question the learned latent w itself stated that elements in w are disentangled from each other, so is the intension behind why we introduce another set $w'$ that are also disentangled.
>
> **Response #5**:
> * Yes, the assumption always holds in our setting that as long as any independent variables in $w$ contribute to a certain property $y_i$ via $w_i'$, its information will be deterministically aggregated to $w_i'$ via Eq. 5 and Eq. 6. Each $w_i'$ is not independent but the set of values of $w$ that $w_i'$ aggregates are independent (line 151-154).  The Eq. 2 holds since properties y are independent conditional on $w’$ and $w’$ aggregates all information from $w: log p(y\vert w)=p(y\vert w’)=\sum_{i=1}^{m}p(y_i\vert w’)=\sum_{i=1}^{m}p(y_i\vert w_i')$. We have added this detailed derivation in Appendix Section A (line 25-28).
> * $W_i’$ are not independent (not disentangled) with each other but variables in w are independent and disentangled with each other. If $y_i$ and $y_j$ are correlated then $w_i’$ is also correlated with $w_j’$ but they share common variables in w. For instance, as shown in Figure 5, w4 is shared by both “x position” and “x+y position” so that we can control those two variables via w4. The $w’$ that predicts “x+y position” aggregates information from w3, w4 and w5 (according to horizontal vector of “x+y position”).

---

> > ### Author Response · Authors · 2022-08-02
> > **Response to Reviewer3 (P6bk) (2/2)**
> >
> > **Comment #6**: The proposed idea is very interesting, however, for property-controlled generation, each time it needs to solve the optimization problem to find the proper w for the target property, this could be expensive, also the paper did not compare with the approaches that do bayesian optimization in the latent to find a best patent variable that can generate the data with the target properties.
> >
> > **Response #6**:
> > * The basic time expense is incurred by the nature of the correlated-property-controllable data generation problem instead of our method. Specifically, correlated property control is a multiobjective optimization problem where different objectives typically have conflicts and hence an optimization is incurred.
> > * Our method does not need to optimize each time. Specifically, once the targeted property is specified, then we will execute the multi-objective optimization to identify the w values. Then we can just randomly sample the z variables' values, together with the identified w, to generate as many as possible new data objects without the need for optimization. This is one of the nice advantages of our method.
> > * We are working on adding one more variant of CorrVAE (CorrVAE-3) based on Bayesian optimization to compare with other comparison models. We will update our results in the paper within the discussion period due to the time limit of the author response period.
> > * The framework based on Bayesian optimization cannot capture the correlation among properties so that it lacks interpretability compared with ours which is able to capture and indicate correlation among properties using the learned mask matrix (Figure 5).

---

> > > ### Author Response · Authors · 2022-08-09
> > > **Update Response #6**
> > >
> > > We sincerely thank the reviewer for the suggestion to compare with Bayesian optimization-based model and the patience. We have conducted additional experiments by predicting properties with the whole $w$ and performing property control via Bayesian optimization. In this case $w’$ and the mask layer are dropped. The results that compare CorrVAE and the Bayesian optimization-based model (BO) are shown in the table below, and are updated as Appendix Table 8 in the paper.
> > >
> > > | Model   | dSprites |              | Pendulum       |                 |
> > > |---------|----------|--------------|----------------|-----------------|
> > > |         | size     | x+y position | light position | shadow position |
> > > | BO      | 0.0033   | 0.0062       | 19.2387        | 17.2858         |
> > > | CorrVAE | 0.0016   | 0.0066       | 15.3900        | 6.0250          |
> > >
> > > Based on the results, CorrVAE achieves smaller MSE on both light position and shadow position of Pendulum dataset. Specifically, for light position, MSE achieved by CorrVAE is 15.3900 which is much smaller than 19.2387 obtained from the Bayesian optimization-based model. For shadow position, MSE achieved by CorrVAE is 6.0250 which is much smaller than 17.2858 obtained from the Bayesian optimization-based model. Besides, on dSprites dataset, CorrVAE achieves the MSE of 0.0016 for the size which is smaller than 0.0033 obtained from the Bayesian optimization-based model. In addition, CorrVAE achieves comparable results on x+y position with the Bayesian optimization-based model. The results indicate that CorrVAE has better performance than the Bayesian optimization-based model on controlling independent variables (i.e., size in dSprites, light position in Pendulum) and correlated properties (shadow position in Pendulum).
> > >
> > > We will add detailed discussion in the paper.

---

> ### Author Response · Authors · 2022-08-02
> **Gemeral response to Reviewer3 (P6bk)**
>
> Thank you very much for your detailed summarization and insightful comments. Please find our answers to your comments/questions below. We have updated our paper based on your suggestions. The summary of updates in the paper are listed in a separate comment on top of the webpage. If you have any further comments/suggestions on the updated version of our paper, we will be glad to improve on them. We also sincerely hope that the revised version and responses could help with an increase in the score.

---

### Official Review · Reviewer_GPtJ · 2022-07-11

**Rating:** 6
**Confidence:** 4
**Soundness:** 1 poor
**Presentation:** 3 good
**Contribution:** 1 poor

**Summary:**

The paper tackles the problem of controllable data generation when the controlled attributes y are correlated.
To do so, it proposes to separate the latent code in a disentangled VAE into two parts, z and w, where w contains the variables that are correlated to y, and z the variables independent of w (and hence to y). The disentangled variables w are combined into correlated variables w' via a learned mask M that indicates for each value of w', which values of w contribute to it. The values of w' are mapped one-to-one to the attributes y using an invertible network.
The model is used for controlled generation, to identify correlations between attributes, and for generation using multi-objective constrained optimization.

**Questions:**

-  The samples shown in Fig3 and Fig4 for dShapes are very bad, even in disentanglement VAE models standard. The shape attribute, that should be encoded in the independent z variables, are not only not conserved with attribute manipulation, but also they do not seem to be shapes that are in the dataset. This drastically limits the usefulness and significance of the method for data generation. Can the authors provide reconstruction errors and FID for generated data, for both seen and unseen combination of attributes?
- The authors discuss in section 5.4 and show in Fig 3 latent traversals of w. However, it would appear that the w space is not that interesting, as they do not align well with w' and y. For instance, both the "x position" and "x+y position" attributes are diluted into two variables (figure 5). This is especially surprising since it increases the KL term in Eq3, and the mask sparcity loss. Meanwhile, the arguably more interesting variable w' that would be used for attribute manipulation is barely investigated. Can the authors provide qualitative samples for latent traversal of w', and quantitative figures on how well the attributes are retained?
- I would also like to see the full mask M in Figure 5. What about "y position" for instance?
- While the authors claim that the ablation CorrVAE-1 that use ground truth masks are achieving better performance than CorrVAE, it is not clear in the result Table 1. This could also indicate that the model is not working as expected and that different variables in the model do not exactly capture the information they are intended to get. Why would CorrVAE-1 would be slightly worse than CorrVAE on some task? How significant is the difference?
- Important implementation details are in Appendix C. At the very least, the fact that the mask is encouraged to be sparse should be mentioned in the main paper. I would argue that approximations and relaxation of the problem, such as Monte-Carlo, Gumbel-SoftMax and Spectral Normalization should also be mentioned in the main paper when used as they are not exact implementations of the provided formula.

Overall, I believe the paper overclaims what is the proposed model able to do. It indeed preserve the controlled attributes, arguably better than the baselines, but seemingly at the cost of loosing the other attributes, even some as fundamental as shape or orientation.
I could change my opinion if the authors can provide evidence that this assessment is wrong (by answering the questions for instance), or if they can show that the trade-off is a desirable feature.


Minors typos:

l199: thrid

l183 w_i^T ·  w_j  : w_i

l326-327: it seem it should be CorrVAE-2 insteand of CorrVAE-1?

Vertical spacing after subsection titles are very unusual.

**Limitations:**

The paper do not mention its limitations.
I believe the weaknesses raised above do qualify as limitations in term of data generation and should be adressed in the paper.

**Strengths And Weaknesses:**

Strengths:
- The paper address the problem of attributes correlation that is often ignored in controlled generation in a principled way.
- Experiments have been conducted on real molecular data, in addition to synthetic images.

Weaknesses:
- The experiments fail to sufficiently back the claims made in the paper. Crucially, the paper is framed as a data generation method but the proposed experimental protocol do not assess the quality of the generated data, only the perservation of a few attributes.

---

> ### Author Response · Authors · 2022-08-02
> **Response to Reviewer2 (GPtJ) (1/5)**
>
> Thank you very much for your detailed summarization and insightful comments. Please find our answers to your comments/questions below. We have updated our paper based on your suggestions. The summary of updates in the paper are listed in a separate comment on top of the webpage. If you have any further comments/suggestions on the updated version of our paper, we will be glad to improve on them. We also sincerely hope that the revised version and responses could help with an increase in the score.
>
> **A summary of updates based on comments from Reviewer2**:\
> (1) We have added experiments to evaluate the quality of generated images using negative log likelihood, reconstruction error and FID values in Appendix Table 5.\
> (2) We have visualized the generated images from CorrVAE in Appendix Figure 1.\
> (3) We have conducted new experiments by controlling five properties: “shape”, “size”, “x position”, “y position” and “x+y position”. The corresponding mask matrix is shown in Figure 5.\
> (4) We added experiments by traversing five latent variables in w that correspond to five properties, respectively, and the change of generated images are visualized in Figure 3.\
> (5) We updated the constraints for multi-objective optimization by controlling “shape”. The generated images are updated and presented in Figure 4.\
> (6) We showcased in Appendix Figure 2 a batch of images generated under four sets of constraints under the multi-objective optimization framework corresponding to Figure 4. Images generated under the same constraint look similar indicating the consistency and replicability of our model.\
> (7) We also added corresponding discussions according above newly added contents.\
>
> **Comment #1**: The experiments fail to sufficiently back the claims made in the paper. Crucially, the paper is framed as a data generation method but the proposed experimental protocol do not assess the quality of the generated data, only the preservation of a few attributes.\
> **Response #1**: \
> Thanks for the comment. We have added the requested evaluation of the data generation performance in addition to attribute preservation in the revised paper, including:\
> (1) For the task of molecule generation on QM9 and QAC datasets, we have added the evaluation on qualities of generated data are measured by validity, novelty and uniqueness, which follows the typical molecule generation works [1] [2], as shown below and in Appendix Table 4.
>
> | Method   | QAC    |         |            | QM9    |         |            |
> |----------|--------|---------|------------|--------|---------|------------|
> |          | Validy | Novelty | Uniqueness | Validy | Novelty | Uniqueness |
> | Semi-VAE | 100%   | 100%    | 37.5%      | 100%   | 100%    | 82.5%      |
> | PCVAE    | 100%   | 100%    | 89.2%      | 100%   | 99.6%   | 92.2%      |
> | CorrVAE  | 100%   | 100%    | 44.5%      | 100%   | 91.2%   | 23.8%      |
>
> (2) For the task of image generation, we have additionally visualized the generated data in Appendix Figure 1. In addition, we added the results regarding FID, reconstruction error and negative log likelihood to evaluate the quality of generated images, which follows the typical image generation works [3] [4]. The results are added in Appendix Table 5. We also have added the relevant discussion in line 339-342, as shown below:
>
> “According to Appendix Table 4 and Appendix Table 5, both molecule and image data are generated well by the proposed model since CorrVAE achieves 100% validity and novelty on molecular generation and comparable reconstruction error and FID values on image generation”
>
> The CorrVAE has superior performance in both negative log likelihood and FID values whereas CorrVAE achieves comparable reconstruction error with CSVAE and PCVAE but performs better than Semi-VAE.
>
> | Method   | -log Prob | Rec. Error | FID   |
> |----------|-----------|------------|-------|
> | CSVAE    | 0.26      | 227        | 86.14 |
> | Semi-VAE | 0.23      | 239        | 86.05 |
> | PCVAE    | 0.23      | 222        | 85.45 |
> | CorrVAE  | 0.22      | 229        | 85.17 |
>
>
> \
> \
> [1] Samanta, Bidisha, et al. "Nevae: A deep generative model for molecular graphs." Journal of machine learning research. 2020 Apr; 21 (114): 1-33 (2020).\
> [2] Luo, Youzhi, Keqiang Yan, and Shuiwang Ji. "Graphdf: A discrete flow model for molecular graph generation." International Conference on Machine Learning. PMLR, 2021.\
> [3] Ak, Kenan E., et al. "Incorporating reinforced adversarial learning in autoregressive image generation." European Conference on Computer Vision. Springer, Cham, 2020.\
> [4] Razavi, Ali, Aaron Van den Oord, and Oriol Vinyals. "Generating diverse high-fidelity images with vq-vae-2." Advances in neural information processing systems 32 (2019).

---

> > ### Comment · Reviewer_GPtJ · 2022-08-05
> > **Discussion**
> >
> > I'd like to thank the authors for the very substancial rebuttal.
> >
> > I believe my concerns have been adequatly adressed. Crucially, the answers corrected a miss-conception from my part as to the goals and motivations of the paper. The additional figures completed the previous results nicely, providing more information about the different components of the model.
> >
> > I still have a minor comment regarding response #3:
> > *(2) The fact that "x position" and "x+y position" attributes are diluted into two variables indicates “x position” and “x+y position” are not 100% colinear and are also not 100% independent. This is why we see each of them has two variables in w, where one variable is shared between them while the remaining one of each of them is not shared.*
> >
> > I agree with this statement. What I meant that it would be very intuitive to encode "x position" as one value $w_i$, and "x+y position" as a pair of value that includes $w_i$ (or vice-versa). Wouldn't this configuration be better in terms of loss?

---

> > > ### Author Response · Authors · 2022-08-05
> > > **Further responses on Comment #3**
> > >
> > > Thanks for your question and our further clarification is as follows.
> > >
> > > In our objective we do have loss (i.e., L1 norm regularization) that encourages to select fewer variables from $w$ for each $w’$. But in the meanwhile we also have other losses that measure the difference between the training data and generated data. The optimization needs to achieve a trade-off among all the losses. We agree that “one variable $w_i$ to control x while a pair of variables that include $w_i$ to control x+y” will theoretically have a lower L1 norm regularization loss. However, this theoretical and perfect situation happens only when the image has no noise and the encoder of variational auto-encoder perfectly-precisely identifies the shapes and calculates out the ground-truth center positions of all the shapes.  But in practice, both image with no noise and perfect encoder doing perfect object encoding for its detection are very hard to achieve.  As a result, the model needs to work with noise related to x and x+y. Hence, one additional variable for x is learned to capture the additional difference between x and x+y caused by the noise, and thus helps decrease the loss that measures the difference between the training data and generated data.

---

> > > > ### Comment · Reviewer_GPtJ · 2022-08-08
> > > > **reply**
> > > >
> > > > I understand the argument, but on the other hand, results obtained by beta-vae [a] and subsequent works [b,c] tends to indicate that it is not so difficult to encode pos_x and pos_y into a single value each without supervision at all.
> > > >
> > > > But this is only a minor concern.
> > > > After reading through the submission again to reassess it, I'm left with a question: how do we properly evaluate the task?
> > > > Indeed, the authors suggest that they are happy with properties that are *changeable, diverse, or even unseen before in training data*. But certainly, they still want the generated properties to be somewhat close to the training data, or else they would be no point to a data-driven approach at all.
> > > >
> > > > An other way to look at it is that, some constrains are hard-set by the experimenter (positions in the example), and others (like shapes) are soft constrains influenced by the dataset and the deep model.
> > > > Given that the later are allowed to some degree to be unseen from the training data, how can we tell if a given level of adherence to the dataset in the generated samples is a desirable feature or an hindrance?
> > > > It seems to me that to evaluate those methods, we would need either an actual downstream task, or that the model provides a control over the degree of adherence to the dataset.
> > > >
> > > > I realize that this comes very late in the discussion process, but I would appreciate if the authors could comment on these views.
> > > >
> > > > In any case, I also understand that this potential problem is inherent to the definition of the task and not easily solved.
> > > > In the meantime, I updated my rating from 3 to 6.
> > > >
> > > > ---
> > > > [a] beta-VAE: Learning Basic Visual Concepts with a Constrained Variational Framework. Irina Higgins et al. ICLR 2017.
> > > >
> > > > [b] Isolating Sources of Disentanglement in Variational Autoencoders. Tian Qi Chen et al.  NeurIPS 2018.
> > > >
> > > > [c] Disentangling by Factorising. Hyunjik Kim and Andriy Mnih. ICML 2018.

---

> > > > > ### Author Response · Authors · 2022-08-09
> > > > > **Further discussion on evaluating generated data**
> > > > >
> > > > > We sincerely thank the reviewer for approving our clarification and we are glad to further discuss the evaluation of generated data.
> > > > >
> > > > > Evaluation metrics include novelty, uniqueness, diversity, validity and similarity to original distribution such as Fréchet Inception Distance (FID). There are some trade-off among them. For example, a totally random data generator could have a very good novelty, while a data copier can follow the training distribution and maintain validity perfectly, but typically neither of them is considered as good enough. More commonly we want a trade-off between these two extremes and we may prefer different ways of the trade-off for different applications. Sometimes we have some implicit criteria (e.g., the generated shape needs to be convex but we are fine if it has a new orientation) in our human mind but the model may not always guarantee this for free unless we “tell” these criteria explicitly to the model (e.g., by adding constraints or property controlling). Also, we agree that generated properties should be somewhat close to the training data, and the “close” here could be “similar to a specific training sample in some aspect but not the others” or “similar to the hybrid of several training samples” or more generally “similarity between training data distribution and generated data distribution”. For example, when our training set contains the shapes like square, circle, and oval, then the generated one could look like a square, circle, oval, or a hybrid of them, or at least should be a convex shape with high probability, and very unlikely to be a bar shape because its corresponding probability to be sampled is low and far away from the “mean” we learned for this generative model.

---

> ### Author Response · Authors · 2022-08-02
> **Response to Reviewer2 (GPtJ) (2/5)**
>
>
> **Comment #2**: The samples shown in Fig3 and Fig4 for dShapes are very bad, even in disentanglement VAE models standard. The shape attribute, that should be encoded in the independent z variables, are not only not conserved with attribute manipulation, but also they do not seem to be shapes that are in the dataset. This drastically limits the usefulness and significance of the method for data generation. Can the authors provide reconstruction errors and FID for generated data, for both seen and unseen combination of attributes?\
> **Response #2**:\
> (1) It is reasonable and our intention to not preserve “shape” attribute. In controllable data generation, our intention is to preserve the controlled properties while making other properties changeable, diverse, or even unseen before in training data. For example, in controllable drug molecule designing, we have two purposes: 1) make sure the generated molecules stick to the required properties of interest. 2) encourage the diversity and novelty of the molecules by varying all the other uncontrolled properties in order to achieve novel drug discovery. Similarly, when generating object images, suppose our controlled properties of interest are only objects' size and position, then analogously we want all the other properties (e.g., shapes and orientations) to be changeable and we'd love to see they are different from the training data (e.g., shapes and orientations unseen before in training data). In all, our paper achieves both 1) preserving controlled properties and 2) perturbating the uncontrolled properties, both of which are highly desired.\
> (2) In addition, to show the versatility of our method, we have added new experiments where we can preserve the shape, by additionally treating "shape" as controlled property. Specifically, in Figure 3 (a) of the main paper, we have shown that when we control “shape” via w and mask, the shape of the objects changes when traversing the corresponding variable in w that controls it. Meanwhile, in Figure 4 of the main paper, we have shown that “shape” can also be well controlled by the constraints of the multi-objective optimization framework. We also visualized the whole batch of images in Appendix Figure 2 generated based on each property constraint of Figure 4 to show that our experiments are consistent and replicable. The added discussion is shown below:
> \
> \
> Line 373-375: “Based on the mask matrix shown in Figure 5, as shown in Figure 3 (a), we traverse the value of w1 within [−5, 5] and the shape of the pattern changes accordingly from ellipse to square.”
> Line 390-405: “The experiments are performed based on the model that controls five properties, shape, scale, x position, y position and x+y position…All properties including shape are roughly aligned with the constraints”
> \
> \
> (3) We have added FID values, reconstruction error and negative log likelihood in Appendix Table 5 to quantitatively evaluate the quality of the shape of generated images:
>
> | Method   | -log Prob | Rec. Error | FID   |
> |----------|-----------|------------|-------|
> | CSVAE    | 0.26      | 227        | 86.14 |
> | Semi-VAE | 0.23      | 239        | 86.05 |
> | PCVAE    | 0.23      | 222        | 85.45 |
> | CorrVAE  | 0.22      | 229        | 85.17 |
>
> The relevant discussion is also added in line 339-342, as shown below:
>
> “According to Appendix Table 4 and Appendix Table 5, both molecule and image data are generated well by the proposed model since CorrVAE achieves 100\% validity and novelty on molecular generation and comparable reconstruction error and FID values on image generation”
> \
> \
> (4) In addition, we also qualitatively evaluate the shape of generated images by visualizing the whole batch of images generated in Appendix Figure 21, based on each property constraint from Figure 4 . The discussion about Appendix Figure 2 is added in line 385-387, as shown below:
> \
> \
> “We also showcased the whole batch (eight) of generated images in Appendix Figure 2 corresponding to each constraint of Figure= 4. All images for the same constraint look similar, indicating the consistency and the replicability of our model.”

---

> ### Author Response · Authors · 2022-08-02
> **Response to Reviewer2 (GPtJ) (3/5)**
>
> **Comment #3**: The authors discuss in section 5.4 and show in Fig 3 latent traversals of w. However, it would appear that the w space is not that interesting, as they do not align well with w' and y. For instance, both the "x position" and "x+y position" attributes are diluted into two variables (figure 5). This is especially surprising since it increases the KL term in Eq3, and the mask sparcity loss. Meanwhile, the arguably more interesting variable w' that would be used for attribute manipulation is barely investigated. Can the authors provide qualitative samples for latent traversal of w', and quantitative figures on how well the attributes are retained?\
> **Response #3**: \
> (1) The w space is interesting in that it contains independent latent variables to aggregate to w’ to control the corresponding y. If there exists one variable in w that contributes to two properties only when those two properties are correlated with each other.\
> (2) The fact that "x position" and "x+y position" attributes are diluted into two variables indicates “x position” and “x+y position” are not 100% colinear and are also not 100% independent. This is why we see each of them has two variables in w, where one variable is shared between them while the remaining one of each of them is not shared. This exactly reflects the rationale of the controllability of our model. In addition, this will not increase the KL term in Eq.3. The KL term is not relevant to the mask matrix, which aggregates information from w to w’, but aims to approximate the posterior of $q_{\phi}(w, z|x)$. Instead, well captured correlation among properties will decrease Eq. 3 and compensate for the possible increase of the mask sparsity loss (L1 norm) in the objective function since the second term of Eq. 3 corresponds to the negative log likelihood of predicted properties. \
> (3) We have added experiments in Appendix Figure 6 to qualitatively visualize the latent traversal of w'. We can see that the property will change accordingly as we traverse the corresponding w’. The attributes are annotated at the right top corner of each generated image. We also added relevant discussions in Appendix line 98-105 as:
> \
> \
> “Moreover, we also traverse the latent variables in $w'$ by simultaneously traversing on latent variables in $w$ corresponding to the associated $w'$ and visualize how the relevant property changes in Appendix Figure 6 . As is shown in Appendix  Figure 6 (a), the shape of the pattern changes from ellipse to square as we traverse on $w_1'$. In Appendix Figure 6 (b), the size of the pattern shrinks as we traverse on $w_2'$. In Appendix Figure 6 (c), the \emph{x position} of the pattern moves from left to right as we traverse on $w_3'$. In Appendix Figure 6 (d), the y position of the pattern moves from top to bottom as we traverse on $w_4'$. In Appendix Figure 6 (e), the x position, y position and x+y position of the pattern simultaneously change as we traverse on $w_4'$, where x position moves from left to right, y position moves from bottom to top and x+y position increase.”
> \
> \
> **Comment #4**: I would also like to see the full mask M in Figure 5. What about "y position" for instance?\
> **Response #4**: \
> (1) The mask M in Figure 5 in the original version is the full  mask regarding three target properties of interest (i.e., x position, x+y position, and size). \
> (2) In addition, by following your suggestion to explicitly control “y position”,  we enrich our training data by adding supervision labels of “y” position so we have new mask. The results of this additional experiments are shown below.:
> \
> \
> (a) We have replaced Figure 5 with the newly learned mask (The old Figure 5 has been moved to Appendix Figure 4). Figure 5 shows that the newly learned mask matrix well captures the correlation among “x position”, “y position” and “x+y position”, where w3 simultaneously controls “x position”, “y position” and “x+y position”, w4 simultaneously controls “x position” and “x+y position” while w5 simultaneously controls “y position” and “x+y position”.
> \
> \
> (b)To validate that the correlation among properties is captured by the learned mask, we visualize generated images by traversing latent variables in w to control corresponding properties according to the mask in Figure 5. We have also updated the relevant discussions in the main paper (line 381-385), as shown below.
> \
> \
> “We also evaluate the more complex setting by traversing the value of w3 within [−5, 5] that simultaneously controls x position, y position and x+y position. Not surprisingly, the position of the pattern changes in both horizontal and vertical directions, corresponding to x+y position. At the mean time, x position and y position change accordingly, as shown in Figure 3 (e).”

---

> ### Author Response · Authors · 2022-08-02
> **Response to Reviewer2 (GPtJ) (4/5)**
>
> **Comments #5**: While the authors claim that the ablation CorrVAE-1 that use ground truth masks are achieving better performance than CorrVAE, it is not clear in the result Table 1. This could also indicate that the model is not working as expected and that different variables in the model do not exactly capture the information they are intended to get. Why would CorrVAE-1 would be slightly worse than CorrVAE on some task? How significant is the difference?\
> **Response #5**: \
> For correlated properties, CorrVAE-1 did outperform CorrVAE, which is what our technique designed for and hence demonstrated our effectiveness. For independent properties case, both CorrVAE-1 and CorrVAE perform similarly which indicate that the ground truth mask does not help a lot for such simple situation. More specifically:\
> (1) For correlated properties, CorrVAE-1 did achieve better performance, such as “x position” and “x+y position” for dSprites dataset and “shadow length” and “light position” for Pendulum dataset, which is aligned with the results in Table 1. For example, MSE of “x position” is 0.0059 for CorrVAE-1, which is better than 0.0077 achieved by CorrVAE. MSE of “x+y position” is 0.0023 for CorrVAE-1, which is better than 0.0066 achieved by CorrVAE. The MSE of “shadow length” and “light position” is respectively 2.4626 and 11.2878 for CorrVAE-1 compared with 10.26 and 15.39 for CorrVAE. \
> (2) For independent properties, CorrVAE-1 would be slightly worse but comparable with CorrVAE since CorrVAE and CorrVAE-1 share the same technique in capturing the independence among properties. For instance, CorrVAE-1 and CorrVAE have comparable results in controlling “size” with value 0.0024 and 0.0016, respectively,  as shown in Table 1. Similarly, for pendulum angle and shadow position of pendulum dataset, CorrVAE and CorrVAE-1 also have comparable results with the value 39.9255 and 36.3700 respectively for pendulum angle, and 6.3579 and 6.0250 respectively for shadow position, as shown in Table 1 We have added the corresponding discussion in the main paper line 325-346.\
> (3) While for the “slightly worse” issue for independent properties, this is because the enforced  supervision of “ground-truth” mask distracts the focus of the model a bit from  “learning independent properties” from “learning corrected properties”. Fortunately, the decrease for independent properties of CorrVAE is small enough to ignore.\
> **Comments #6**: Important implementation details are in Appendix C. At the very least, the fact that the mask is encouraged to be sparse should be mentioned in the main paper. I would argue that approximations and relaxation of the problem, such as Monte-Carlo, Gumbel-SoftMax and Spectral Normalization should also be mentioned in the main paper when used as they are not exact implementations of the provided formula.\
> **Response #6**: \
> Thank you for this valuable suggestion. We have added the detailed usage and implementation of the mask matrix to the main paper (line 170- 172)\

---

> ### Author Response · Authors · 2022-08-02
> **Response to Reviewer2 (GPtJ) (5/5)**
>
> **Comments #7**: Overall, I believe the paper overclaims what is the proposed model able to do. It indeed preserve the controlled attributes, arguably better than the baselines, but seemingly at the cost of loosing the other attributes, even some as fundamental as shape or orientation. I could change my opinion if the authors can provide evidence that this assessment is wrong (by answering the questions for instance), or if they can show that the trade-off is a desirable feature.\
> **Response #7**: \
> Thanks for the opportunity for us to clarify. We have added the clarification and new experiments as introduced in the following:\
> (1) In controllable data generation, our intention is to preserve the controlled properties while making other properties changeable, diverse, or even unseen before in training data. For example, in controllable drug molecule designing, we have two purposes: 1) make sure the generated molecules stick to the required properties of interest. 2) encourage the diversity and novelty of the molecules by varying all the other uncontrolled properties in order to achieve novel drug discovery. Similarly, when generating object images, suppose our controlled properties of interest are only objects' size and position, then analogously we want all the other properties (e.g., shapes and orientations) to be changeable and we'd love to see they are different from the training data (e.g., shapes and orientations unseen before in training data). In all, our paper achieves both 1) preserving controlled properties and 2) perturbating the uncontrolled properties, both of which are highly desired.\
> (2) In addition, to show the versatility of our method, we have added new experiments where we can preserve the shape, by treating "shape" as controlled property. Specifically, in Figure 3 (a) of the main paper, we have shown that when we control “shape” via w and mask, the shape of the objects changes when traversing the corresponding variable in w that controls it. Meanwhile, in Figure 4 of the main paper, we have shown that “shape” can also be well controlled by the constraints of the multi-objective optimization framework. We also visualized the whole batch of images in Appendix Figure 2 generated based on each property constraint of Figure 4 to show that our experiments are consistent and replicable.\
> **Comments #8**: Minors typos: l199: thrid l183 w_i^T · w_j : w_i l326-327: it seem it should be CorrVAE-2 insteand of CorrVAE-1? Vertical spacing after subsection titles are very unusual.\
> **Response #8**:\
> Thank you for pointing out these typos. We have corrected all of them in the revised paper

---

### Official Review · Reviewer_2J2q · 2022-07-11

**Rating:** 8
**Confidence:** 5
**Soundness:** 3 good
**Presentation:** 3 good
**Contribution:** 4 excellent

**Summary:**

Generating data with multiple constraints on its correlated properties is a critical task. The authors address this task by designing a new mask pooling layer to identify and control correlated properties using independent latent variables. These latent variables are bond to properties based on the mutual dependence. Then these latent variables are optimized to generate data with desired properties under a multi-objective optimization framework. The effectiveness of the proposed model has been shown on the molecule and image datasets in the evaluation session.

**Questions:**

1. Provide more discussion about the results in Table 1. Why CorrVAE has worse performance than CSVAE on two independent variables x position and scale?
2. Discuss the missing reference [1] in the Related works session.


**Limitations:**

The paper discusses the limitation when pointing out the potential future works in the Conclusion.
That the work does not have potential negative social impact.

**Strengths And Weaknesses:**

Strengths:
1. The tasks approached by the paper are challenging but practically important. The framework proposed by the authors is novel.
1.1 The proposed framework includes a mask pooling layer learned to capture and control correlated properties via the independent latent variables. This also to some extent adds interpretability to the model.
1.2 In addition, the framework employs a set of bridging latent variables w to aggregate information from w to predict properties. The exact recovery of w from properties can be achieved via an invertible constraint of mutual dependence.
1.3 The authors formally propose a multi-objective optimization framework for simultaneously control correlated properties of generated data. It comes with the generality in terms of accommodating different optimization goals and constraints, which looks reasonable and interesting for controllable data generation.

2. The experiments show the effectiveness of the model.
2.1.	The mask pooling layer works well given the results shown in Figure 5 under the setting that x position and x+y position should be correlated with each other.
2.2. Figure 3 shows the effect of disentanglement of the latent variable w. Correlated properties x+y position can be well controlled by the corresponding latent variable given by the results of mask pooling layer.
2.3. As shown in Table 1, CorrVAE can handle correlated properties x+y position better than other comparison models on dSprites dataset. Table 2 shows the effectiveness of CorrVAE on molecule datasets on MolWeight and logP properties.

3 The code of the paper is well packaged and published.

Weakness:
1. In Table 1, although the proposed method dominating the others, it would be great to provide more analysis about the results why sometimes the degree of superiority is very large while sometimes its performance is close to the others.
2. The white text on Figure 3 and Figure 4 can be larger to be more readable. The fond size of Table 1 and Table 2 should be aligned.
3. Missing references: For example [1] below can be discussed in the Related works session.

[1] Xie, Yutong, et al. "Mars: Markov molecular sampling for multi-objective drug discovery." arXiv preprint arXiv:2103.10432 (2021).

---

> ### Author Response · Authors · 2022-08-02
> **Response to Reviewer1 (2J2q)**
>
> We appreciate so much for reviewer's comments and feedbacks that made our paper further improved while we were addressing the concerns. We are also glad that reviewers approved our work regarding important tasks that we solve.
> \
> \
> **Comment #1**: In Table 1, although the proposed method dominating the others, it would be great to provide more analysis about the results why sometimes the degree of superiority is very large while sometimes its performance is close to the others. Why CorrVAE has worse performance than CSVAE on two independent variables x position and scale?\
> **Response #1**:\
> (1) We have refined our discussion of Table 1 by adding more details in the main paper (line 323-342):
> \
> \
> "We evaluate the learning ability of the proposed model and ... CorrVAE-2 models $w$ to $w′$ using simple linear regression, which cannot capture the non-linear correlation among properties that might exist in the dataset".\
> \
> (2) CSVAE models the distribution of $w$ given properties $y$ via a simple Gaussian MLP whereas CorrVAE needs to learn the whole variance-covariance matrix (Eq. (9)) and the the mapping function from $w$ to $w'$ (Eq. (5)) if correlation exists among properties. Hence, CorrVAE needs to learn more parameters than CSVAE. This will lead to a comparable or slightly worse performance given the potential overfitting if the pattern of the dataset is linear and simple to learn (e.g., shape or x position for dSprites dataset). As shown in Figure 1, CorrVAE works much better than CSVAE if more complex settings exist, such as complex patterns of the dataset (e.g., Pendulum) or there are correlation among properties since in this case the correlation contributes to the prediction of properties and the complex data structure needs more parameters to learn. CSVAE cannot well capture the correlation among properties even in the dSprites dataset, which can be validated from the results in Table 1 in that it has a MSE of 0.3563 in predicting x+y postion which is much worse than 0.0066 obtained from CorrVAE.
>
> **Comment #2**: The white text on Figure 3 and Figure 4 can be larger to be more readable. The fond size of Table 1 and Table 2 should be aligned.\
> **Response #2**:\
> Thank you for the valuable suggestions. We have enlarged the font size of our Figure 3 and Figure 4.
>
> **Comment #3**: Missing references: For example [1] below can be discussed in the Related works session.\
> **Response #3**:\
> Thank you for the valuable suggestions. We will surely discussed the reference in our revised paper.

---

> > ### Comment · Reviewer_2J2q · 2022-08-09
> > **Responses**
> >
> > My comments have been properly addressed.

---

### Author Response · Authors · 2022-08-02
**A final summary of updates in the paper**

We sincerely appreciate for reviewers' comments and feedbacks that made our paper further improved when we were addressing their concerns. We are also glad that reviewers approved our clarifications and are satisfied with how we addressed their comments. The following provides a final summary of the updates:

(1) We have added experiments to evaluate the quality of generated images using negative log likelihood, reconstruction error and FID values in Appendix Table 5.\
(2) We have visualized the generated images from CorrVAE in Appendix Figure 1.\
(3) We have conducted new experiments by controlling five properties: “shape”, “size”, “x position”, “y position” and “x+y position”. The corresponding mask matrix is shown in Figure 5.\
(4) We added experiments by traversing five latent variables in w that correspond to five properties, respectively, and the change of generated images are visualized in Figure 3.\
(5) We updated the constraints for multi-objective optimization by controlling “shape”. The generated images are updated and presented in Figure 4.\
(6) We showcased in Appendix Figure 2 a batch of images generated under four sets of constraints under the multi-objective optimization framework corresponding to Figure 4. Images generated under the same constraint look similar indicating the consistency and replicability of our model.\
(7) We also added corresponding discussions according above newly added contents. The location of these discussion will be identified in the response to reviewers.\
(8) All old figures that control three properties (i.g., “size”, “x position” and “x+y position”) are moved to the Appendix: Old Figure 3 is moved to Appendix Figure 3. Old Figure 4 is moved to Appendix Figure 5. Old Figure 5 is moved to Appendix Figure 4.\
(9) We added results that compare CorrVAE and the Bayesian optimization-based model (BO) in Appendix Table 8. We also added relevant discussions in Appendix line 83-95.

---

### Meta-Review · Area_Chair_fnmy · 2022-08-26

**Recommendation:** Accept
**Confidence:** Certain

**Metareview:**

All three reviewers argue to accept (albeit one borderline).

Extremely substantial response from the authors addressing individual reviewer comments, which led to reviewers raising their scores, and to a much revised paper with new experiments. This effectively led to a second round of review, with engaged reviewers who confirmed their concerns have largely been met.

**Award:**

No

---

### Decision · Program_Chairs · 2022-09-14

Accept